# Visualizing RNA conformational and architectural heterogeneity in solution

Jienyu Ding[1], Yun-Tzai Lee [1], Yuba Bhandari[1], Charles D. Schwieters [2], Lixin Fan[3], Ping Yu[1], Sergey G. Tarosov[4], Jason R. Stagno [1], Buyong Ma [5], Ruth Nussinov[5], Alan Rein [6], Jinwei Zhang [7] & Yun-Xing Wang [1] ✉

RNA flexibility is reflected in its heterogeneous conformation. Through direct visualization using atomic force microscopy (AFM) and the adenosylcobalamin riboswitch aptamer domain as an example, we show that a single RNA sequence folds into conformationally and architecturally heterogeneous structures under near-physiological solution conditions. Recapitulated 3D topological structures from AFM molecular surfaces reveal that all conformers share the same secondary structural elements. Only a population-weighted cohort, not any single conformer, including the crystal structure, can account for the ensemble behaviors observed by small-angle X-ray scattering (SAXS). All conformers except one are functionally active in terms of ligand binding. Our findings provide direct visual evidence that the sequence-structure relationship of RNA under physiologically relevant solution conditions is more complex than the one-to-one relationship for well-structured proteins. The direct visualization of conformational and architectural ensembles at the single-molecule level in solution may suggest new approaches to RNA structural analyses.

Ever since the first protein structure was determined in 1958[1], we have become accustomed to the idea that a given primary amino acid sequence folds into corresponding secondary structures and a cognate tertiary structure of a well-folded protein. This idea has been reinforced by the recent, rapid, and exciting developments in predicting protein structures from primary sequences and extensive knowledge available in structural databases[2–4]. For RNA, the functional roles of structural dynamics continue to emerge in a number of biological and cellular contexts[3]. RNA differs from protein in several fundamental aspects that dictate its folding. It is a negatively charged polyelectrolyte, highly hydrophilic, and made of only four different

building blocks that result in a highly degenerate folding-energy landscape. That is, one RNA primary sequence may adopt multiple energetically degenerate secondary, tertiary, or higher-order structures, whose architectures, populations of various conformers, and motion timescales, contribute to the versatile functions of RNA[5]. Indeed, extensive and pioneering studies have revealed that RNA structures are far more dynamic and heterogeneous than what could be described by single conformers[5–8]. Nevertheless, those studies provide only time-averaged behaviors of ensembles, not explicit spatial descriptions about individual molecules or subgroups of conformers in low populations. Moreover, current high-resolution

[1]Protein-Nucleic Acid Interaction Section, Center for Structural Biology, National Cancer Institute, Frederick, MD 21702, USA. [2]Computational Biomolecular Magnetic Resonance Core, National Institute of Diabetes and Digestive and Kidney Diseases, National Institutes of Health, Bethesda, MD 20892, USA. [3]Basic Science Program, Frederick National Laboratory for Cancer Research, Small Angle X-ray Scattering Core Facility of National Cancer Institute, National Institutes of Health, Frederick, MD 21702, USA. [4]Biophysics Resource, Center for Structural Biology, National Cancer Institute, Frederick, MD 21702, USA. [5]Computational Structural Biology Section, Frederick National Laboratory for Cancer Research in the Laboratory of Cancer Immunometabolism, National Cancer Institute, Frederick, MD 21702, USA. [6]Retrovirus Assembly Section, HIV Dynamics and Replication Program, National Cancer Institute, Frederick, MD 21702, USA. [7]Laboratory of Molecular Biology, National Institute of Diabetes and Digestive and Kidney Diseases, Bethesda, MD 20892, USA. ✉e-mail: wangyunx@mail.nih.gov

techniques for structure determination (NMR, crystallography, cryo-EM) rely on signal enhancements over a large number of molecules of the same or very similar conformations to improve the signal-to-noise ratio. These methods are limited to studies of relatively homogeneous samples by driving the molecules to a uniform conformation through extreme buffer conditions (e.g., high $Mg^{2+}$ concentration) or by removing heterogeneous species during or after purification (e.g., denaturing, annealing). In either case, such an artificially selected conformation is insufficient to depict the conformational ensemble that exists in solution, thereby limiting the perception of RNA molecules as static structures. As a result, both the extent and nature of RNA structural heterogeneity under near-physiological solution conditions have remained largely unknown.

The question, therefore, of whether the description "one sequence, one structure" is appropriate for RNA remains to be answered, and becomes even more urgent given the major development in predicting RNA structures[9]. This question is fundamental to the understanding of functional RNA structural dynamics, and the answer requires direct examination of individual molecules in solution. The recent evolution of atomic force microscopy (AFM) enables such direct visualization with several clear advantages[10–18]. Measurements can be performed in physiologically relevant buffer conditions; sample consumption is extremely low, requiring only microliter volumes at nanomolar concentrations; and molecules can be observed in their native states without any manipulation, e.g., labeling, freezing, staining, or crystallization[12]. Moreover, as data are recorded in real space, a high signal-to-noise level can be achieved from a single image. Thus, the solution AFM method is well-suited for studying highly heterogeneous molecular systems under near-native conditions.

Cobalamin riboswitches are one of the most widely distributed riboswitches, regulating B12 biosynthesis in bacteria[19–22]. Chemical probing, analytical size-exclusion chromatography, and multi-angle light scattering (SEC-MALS) data showed that possible heterogeneous conformations for the RNA exist in solution[23–25]. In crystals, the structure of adenosylcobalamin riboswitch aptamer domain from *Thermoanaerobacter tengcongensis* (rCbl), in complex with adenosylcobalamin (AdoCbl), exhibited a compact tertiary fold stabilized by long-range kissing-loop (KL) interactions[26]. Using the rCbl RNA as a proof-of-principle, our solution AFM data show that a single RNA sequence can form multiple conformations and individually definable multimeric architectures under physiologically relevant $Mg^{2+}$ concentration, thereby providing new insights into how RNA structures should be perceived and investigated.

## Results
### Visualizing RNA conformational and architectural heterogeneity
High-resolution AFM images of the rCbl RNA reveal structural heterogeneity among thousands of particles, which are individually classified as monomeric (*Y-shape* (*Y*), *P-shape* (*P*), *candy-shape* (*candy1* and *candy2*), *compact*), dimeric, or multimeric (Fig. 1). In the absence of ligand (2073 particles), the aptamer is almost exclusively monomeric, whereas in the presence of ligand (3410 particles), it is predominantly dimeric (62.7%), suggesting that the dimers are stabilized by ligand binding. Second, the presence of ligand reveals a marked decrease in the populations of *candy* (from 49.4 to 2.3%) and *P* (from 22.5 to 9.1%) monomer conformations, coupled with an increase (from 1.6 to 62.7%) in the dimer population, indicating that those monomer conformations may undergo ligand-induced structural changes that ultimately drive dimer formation. The third major species observed, *Y*, exhibits a very similar population (~22%) under both conditions, suggesting that its structure is independent of ligand and may represent a binding-incompetent or misfolded conformation. It is noteworthy to mention that, with direct visualization by AFM, minor species with only a single or few copies can be detected (Fig. 1a, red arrows), and all conformers

are present, albeit with different populations, under all conditions tested in this study, including temperature, salt concentrations, folding processes, purification procedures, and storage conditions (Supplementary Fig. 1). To confirm that such structural heterogeneity is not the result of using an RNA sequence from a thermophile, which may not sufficiently sample its folding landscape at the in vitro transcription temperature of 37 °C, we followed the same transcription, purification, and imaging procedures using the mesophilic atypical cobalamin riboswitch (aCbl) from *B. subtilis*. The AFM images show that in both the absence and presence of ligand, the conformation of aCbl is heterogeneous, similar to that of rCbl, albeit with different conformer populations, consisting of *candy, Y, P, compact,* and dimers (Supplementary Fig. 2).

### Topological structures of conformers
We further examined the various conformers by recapitulating their 3D topological structures, one for each monomer and dimer class, using coarse-grained dynamic fitting, in which the high-resolution single-particle AFM images were applied as topological constraints. Each class was determined to have a distinct structural fold (Fig. 2 and Supplementary Table 1), while they all share the same secondary structure. The *compact* monomer showing the KL interaction between L5 and L13 most closely resembles the ligand-bound crystal structure (PDB: 4GMA)[26], with a cross-correlation score of ~0.87 (see "Methods") between the AFM image and the crystal structure. Interestingly, however, this crystal-structure-like monomer is present at an extremely low population in both the presence and absence of ligand at 1 mM $Mg^{2+}$ concentration. All other monomer classes reveal more extended conformations, where L13 is disengaged from L5 (Fig. 2). In the cases of *P* and *candy*, the P13 helix becomes an extension of P1. As a result, L5 and L13 are free to form intermolecular KL interactions with other molecules, as observed in the majority of dimer classes (Figs. 2b and 3).

### Heterogeneous conformers account for ensemble behaviors
The presence of multiple conformations in solution under physiological $Mg^{2+}$ concentration is corroborated by SAXS measurements using the same batch of RNA. The topological AFM-derived structures and the volume fraction of each monomer and dimer conformation, in the absence or presence of ligand, were used to synthesize individual SAXS curves, which were then compared to the experimental SAXS data (Fig. 2c, d). None of the back-calculated SAXS curves of a single species, including that of the crystal structure, could be fit to the experimental data, consistent with highly heterogeneous RNA conformations revealed by AFM. The experimental data in both cases could only be approximated using a fractional combination of all monomer and dimer topological structures (Fig. 2c–e and Supplementary Fig. 3c). In the absence of ligand, the populations of various conformers derived from SAXS-ensemble fitting show several differences with respect to the particle tallies from AFM images, particularly in the number of monomeric vs. dimeric species (compare Figs. 1e and 2e). This observation is most likely due to the differences in sample concentrations required for each method. At the low nanomolar concentration for AFM, larger populations of *candy* and *P* are observed, with almost no dimer population, whereas the opposite is true for the SAXS-derived populations under micromolar concentrations. These observations suggest that the monomeric species of *candy* and *P* can readily convert to form dimers in a concentration-dependent manner. In the presence of ligand, the dimeric species make up the dominant populations in both methods. Importantly, the *Y* conformer population remains relatively unchanged in the absence or presence of ligand and is independent of rCbl concentration.

### Heterogeneous conformers are active
The isotherm and thermogram from isothermal titration calorimetry (ITC) data (Fig. 2f and Supplementary Fig. 4) indicate that the reaction

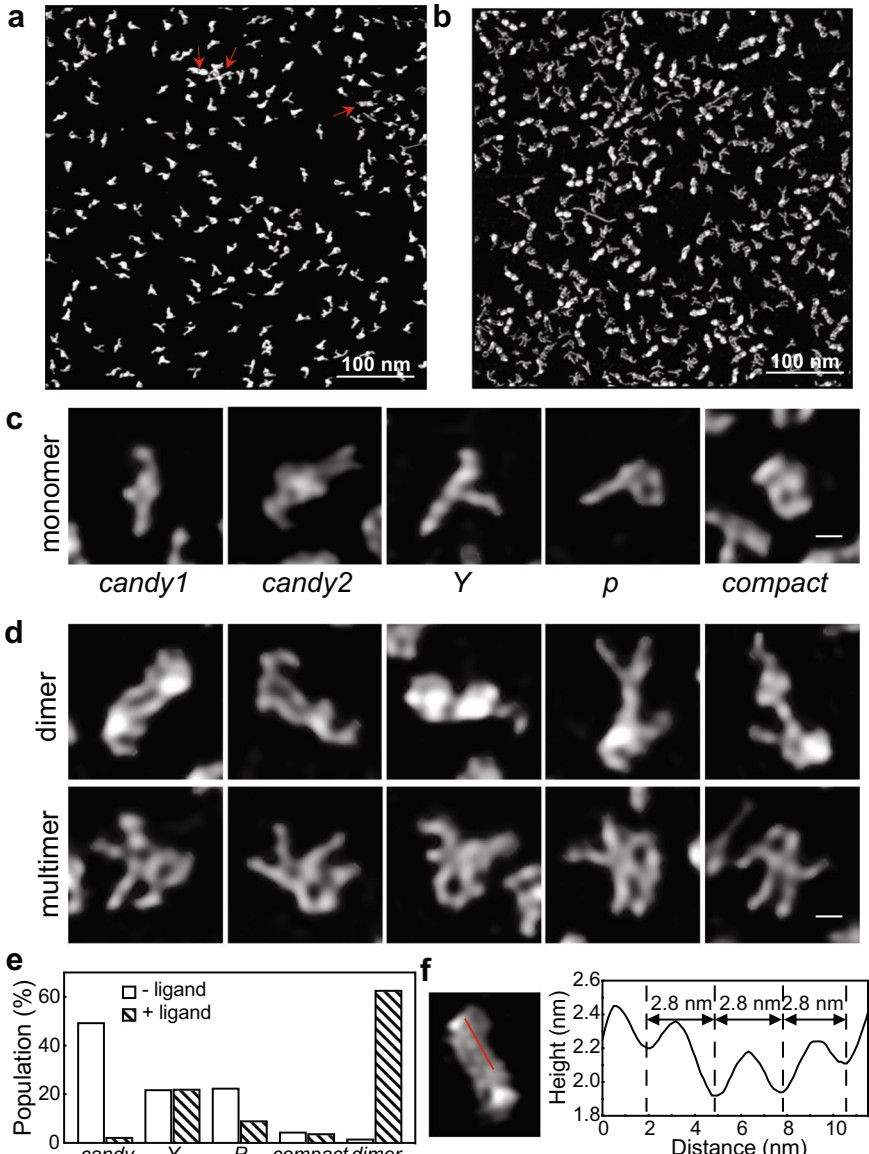

**Fig. 1 | Conformational heterogeneity of rCbl in solution.** Sections of representative high-resolution AFM images of rCbl in the absence (**a**) or presence (**b**) of ligand. Red arrows indicate the detection of minor species down to only one or a few copies in solution. A total of 12 images were recorded for rCbl in the absence of ligand, and 9 images for rCbl in the presence of ligand. **c** Monomeric conformers. **d** Dimers are classified based on shape. The scale bars shown in the right bottom corner of (**c**) and (**d**) are 5 nm. **e** Tallies of individual monomeric conformations (*candy*, *Y*, *P*, *compact*) and all dimeric species, in the absence (2073 particles) or presence (3410 particles) of ligand. **f** Height and distance parameters measured across the surface (red line) of a particle showing periodic corrugation of 2.8 nm, corresponding to RNA helical pitch. Source data are provided as a Source Data file.

between aptamer and ligand does not follow a simple 1:1 binding regime but rather a mixture of endothermic and exothermic events. Singular value decomposition (SVD) analysis reveals that the data are adequately fit with a minimum of two principal components representing interconverting active conformations with different populations (Fig. 2f and Supplementary Table 2). Moreover, the resulting apparent binding stoichiometry (Supplementary Table 2, $N_{app} = 0.78$) reflects that the majority of heterogeneous species are active in terms of ligand binding, and only ~22% of the RNA is incapable of ligand binding, which is consistent with the *Y* conformer population observed by AFM that did not respond to ligand (Fig. 1e). Of the binding-competent aptamers revealed by ITC, the majority (component 1: ~69%) exhibits a typical sigmoidal isotherm signature of an exothermic binding interaction with ligand, which likely involves only minor conformational changes. Component 2 (~19%) reflects an endothermic reaction, most probably associated with large conformational changes

by an induced-fit binding scenario, followed by an exothermic reaction that is similar to component 1. The endothermic nature of this process is consistent with the enthalpic cost required to break existing contacts that stabilize the non-native conformations. The ITC data are consistent with the presence of heterogeneous conformational species and the ligand-binding-incompetent conformer.

### Conserved RNA−RNA interactions drive the formation of architectural conformers

The promiscuity found in the dimer/multimer formations of rCbl is not caused by non-specific interactions, such as random basepairing or electrostatic interactions, which one would otherwise suspect if no direct visual evidence were available. Instead, it is driven by well-known RNA−RNA interaction motifs based on the AFM-derived structural models. Most notable in this case is the KL interaction between L5 and L13 of two molecules, which appears in all but one of the dimer

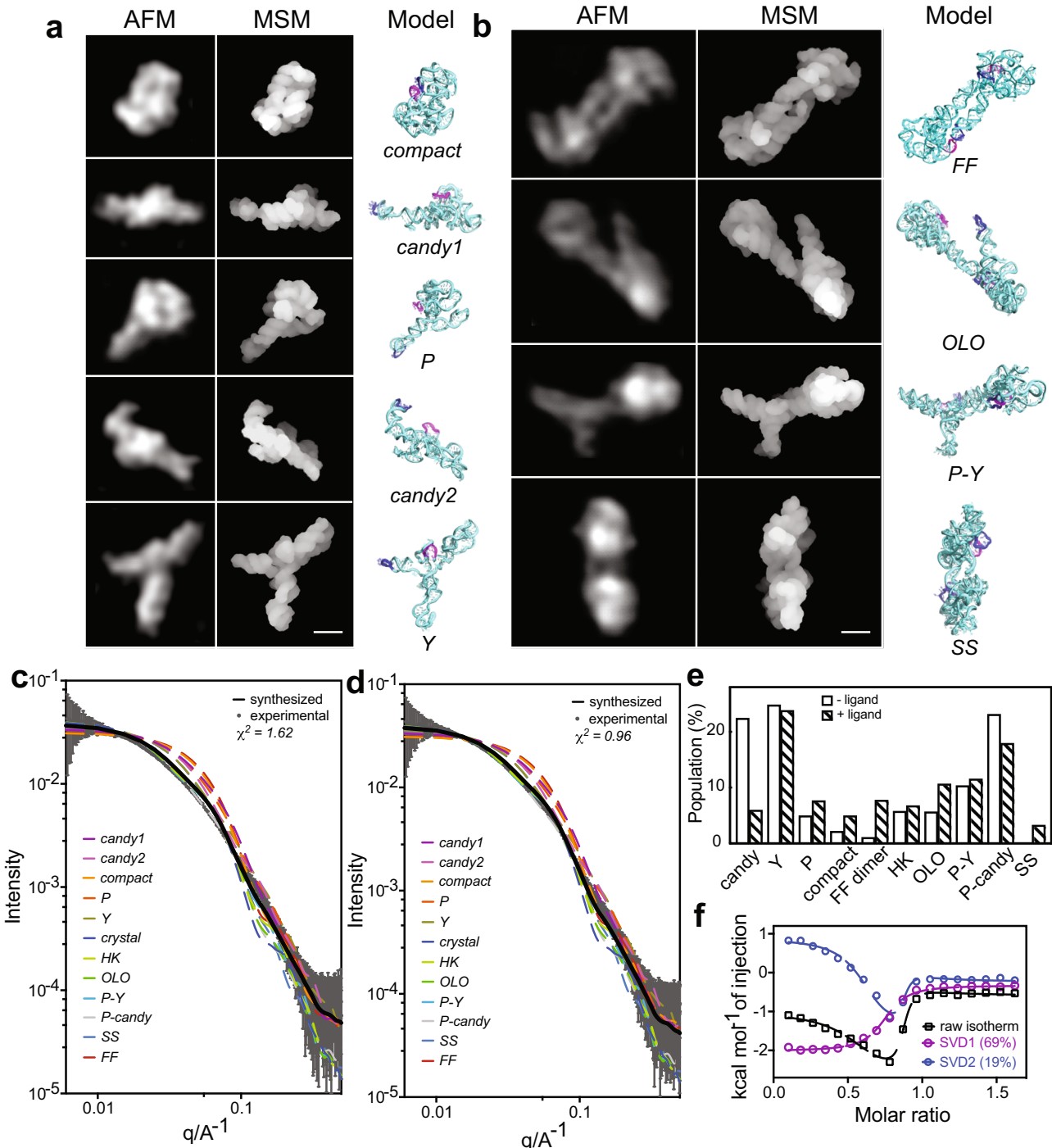

**Fig. 2 | Recapitulated 3D topological structures of the rCbl heterogeneous conformers corroborated by SAXS and ITC. a, b** Individual particles cropped from the experimental AFM images shown in Fig. 1a, b, the recapitulated structures in the molecular surface model (MSM) and cartoon models for the monomeric (**a**) and dimeric (**b**) conformers. L5 and L13 are colored in magenta and blue, respectively. The scale bars of 5 nm are shown in the right bottom corner for (**a**) and (**b**). **c, d** Comparison of the experimental SAXS curve (gray solid line) with back-calculated SAXS curves of the individual conformers (colored dash lines) with the synthesized SAXS curve (black solid line). Experimental SAXS data were recorded for rCbl (1 μM) in the absence (**c**) or presence (**d**) of 10 μM AdoCbl. Data are

presented as mean value +/− error (propagation of uncertainty) as bar and whisker (n = 223 measurements). The synthesized SAXS curves were calculated using the structures of all conformers ("Methods") with $\chi^2 = 1.62$ for rCbl in the absence of ligand and $\chi^2 = 0.96$ for rCbl in the presence of ligand. **e** The conformer population tallies that generate the best fit to the experimental SAXS data. **f** Deconvoluted (SVD1 and SVD2) isotherms of rCbl titrated with AdoCbl (Supplementary Table 2). The eigenvalue fractions for principal component 1 (SVD1) and component 2 (SVD2) are about 69% and 19%, respectively. The raw isotherms are shown in Supplementary Fig. 4a. Source data are provided as a Source Data file.

classes observed by AFM (Figs. 2b and 3). Specifically, this interaction drives the formation of the fully-formed (*FF*), one-leg-out (*OLO*), and hook (*HK*) dimers between two *P*-shape or two *candy*-shape homo-conformers, and *P-Y* and *P-candy* dimers between two hetero-

conformers (Fig. 3). The *OLO* dimer is likely an *FF* dimer in the process of either forming or dissociating. Investigation of the crystal structure of rCbl[26] shows that it may be interpreted as a symmetrical dimer formed through intermolecular KL loop interactions, similar to

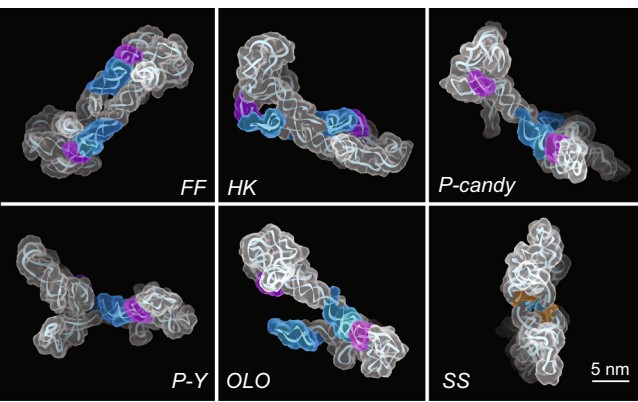

**Fig. 3 | Formation of the rCbl heterogeneous dimers through conserved RNA−RNA interaction motifs.** Five of the six dimer classifications observed by AFM exhibit one or two KL interactions between L5/P5 (purple) and L13/P13 (blue): fully-formed (*FF*), hook (*HK*), *P-candy*, *P-Y*, and one-leg-out (*OLO*). The shoulder-to-shoulder (*SS*) dimer forms through symmetrical tetraloop (cyan)−minor-groove (orange) interactions. The AFM molecular surfaces used for structure calculation were derived from individual particles selected from the AFM image in Fig. 1b.

the dimeric structure reported for aCbl[25] (Supplementary Fig. 5). The observation of the *OLO* dimer provides further evidence supporting the dimerization via the intermolecular KL interaction. Another well-known interaction motif that plays a role in rCbl dimer formation is the tetraloop−minor−groove interaction, as observed in the *SS* dimer (Fig. 3). Common to all these interactions is the requirement of chemical and structural complementarity between interacting partners. In essence, the RNA−RNA interactions are governed by principles similar to protein−protein interactions, which include complementarity of size, shape, chemical composition, and structure[19, 20]. The interactions in RNA and proteins differ only in how specificity is achieved. In RNA, both the chemical and structural complementarity among structural elements, such as KLs, tetraloop−minor−groove interactions, and helical stacking, may increase the promiscuity in forming inter-molecular architectures. In contrast, hydrophobic contact surfaces aided by electrostatic and polar complementarity may drive inter-molecular interactions in proteins.

### Ligand-binding-incompetent *Y*-shape conformers

The topological structures of *Y* conformers, subclassified as *Y1* and *Y2* (Fig. 4b), both exhibit extended and partially folded structures lacking the L5/L13 KL interaction, yet with an intact P6 extension. The P6 extension contributes to one leg of the "*Y*," while the other two legs could be either P2, P13, or P1 together with P13 (Supplementary Fig. 6). To further investigate the correlation between *Y* conformers and inactive species, we repeated AFM and ITC analyses for two mutants of rCbl, in which the essential KL interaction is abrogated by deleting P13 (M2) or by replacing L13 with a tetraloop (M3) (Fig. 4b and Supplementary Figs. 6 and 7). As expected, AFM observation of either of these mutants shows >80% *Y* population, with the remainder of particles resembling *P* or *candy*, each at populations <10%, and no dimers (Fig. 4c and Supplementary Fig. 8). M2 *Y* conformers exhibit a larger mean particle volume, even though both mutant *Y* conformers have similar particle diameters (Supplementary Fig. 9), suggesting that the P13 deletion results in more extended conformations. This is further supported by the topological structures calculated from representative *Y* particles of each mutant (Fig. 4b), which show very similar conformations (also similar to non-mutant *Y* conformers), where the P6 extension constitutes one leg, and the other two legs are comprised of P2 and P4/P5 (M2) or P2 and P13 (M3), respectively (Supplementary Fig. 6). More importantly, as in the case of rCbl, the population of mutant *Y* conformers does not change in the presence of ligand

(Fig. 4c), once again indicating that these *Y* conformers are a ligand-binding-incompetent species. This is further confirmed by ITC results where M2 and M3 mutants show no AdoCbl binding activity (Fig. 4d and Supplementary Fig. 4c, d). The presence of ligand-binding-incompetent conformers was also reported in a close cousin of rCbl, hydroxocobalamin (HyCbl)-binding riboswitch (*env8*HyCbl)[23].

## Discussion

Both RNA and protein can fold into three-dimensional (3D) structures. One of the basic premises of revolutionary protein-structure prediction methods[2–4] is that a primary sequence folds into a specific 3D structure under near-physiological buffer conditions. Such a one-to-one relationship is appropriate for well-folded protein classes, as clearly evidenced in the structure database and demonstrated through applications of AlphaFold[2]. Although these advances in protein-structure prediction provide hope and encouragement for overcoming unique challenges in elucidating RNA 3D structures[9], abundant experimental evidence suggests that a given RNA sequence can adopt diverse structural folds under physiologically relevant conditions[5–8]. Many of these functionally important folds might have been precluded from structure determination using current methods that require homogeneous samples or under non-physiologically relevant conditions. The direct visualization by AFM presented here illustrates the heterogeneity in both conformation and higher-order architecture that can exist for a single RNA sequence. These findings were consistent for both thermophilic (rCbl) and mesophilic (aCbl) cobalamin-binding aptamers in this study. Of note is the propensity of these RNAs to form various dimers at nanomolar concentrations and under physiologically relevant buffer conditions, whose populations increase in the presence of ligand and primarily involve KL interactions. Such intermolecular species do highlight the promiscuity of degenerate RNA−RNA interactions. Our ITC data indicate that all in vitro transcribed RNA particles observed in this study, except for the *Y* conformer, represent functional and active species capable of ligand binding. Principal component analysis of the ITC data indicates a mixture of conformational species, some of which may be capable of binding ligand directly (conformational selection), whereas others likely undergo conformational changes through an induced-fit binding modality.

Being conformational switches, whether co-transcriptional or post-transcriptional, RNA aptamers must be capable of alternate conformations that are intrinsic to their function. Moreover, co-transcriptionally folded aptamers may inevitably form partially folded structures as they are being transcribed. Some of the conformers observed by AFM may indeed represent structures along the folding trajectory but are still active species (i.e., not misfolded) capable of ligand binding through induced-fit conformational changes. Such alternate conformations could be informative for better understanding ligand selectivity by riboswitches capable of binding ligand derivatives with varying affinities. For example, it was shown that the promiscuity of ligand binding by aCbl may be correlated with its structure, namely the organization of peripheral domains[25]. More generally, an aptamer may adopt a more preferred conformation in response to a particular ligand. The use of AFM may aid in providing the structural basis for these differences and their biological implications.

Our findings should stimulate discussion about approaches to RNA preparation and methodologies for structural analysis, particularly with respect to the detection of conformational species. This includes potential implications for data interpretation of structural and biochemical experiments, which are often conducted under milli- to micromolar concentrations, as well as general considerations for understanding RNA structure-function relationships in their native contexts. In particular, $Mg^{2+}$ concentration is known to significantly affect the structure and dynamics of virtually all classes of folded RNAs and long non-coding RNAs, highlighting the importance of studying RNA molecules under physiologically relevant $Mg^{2+}$ conditions and at

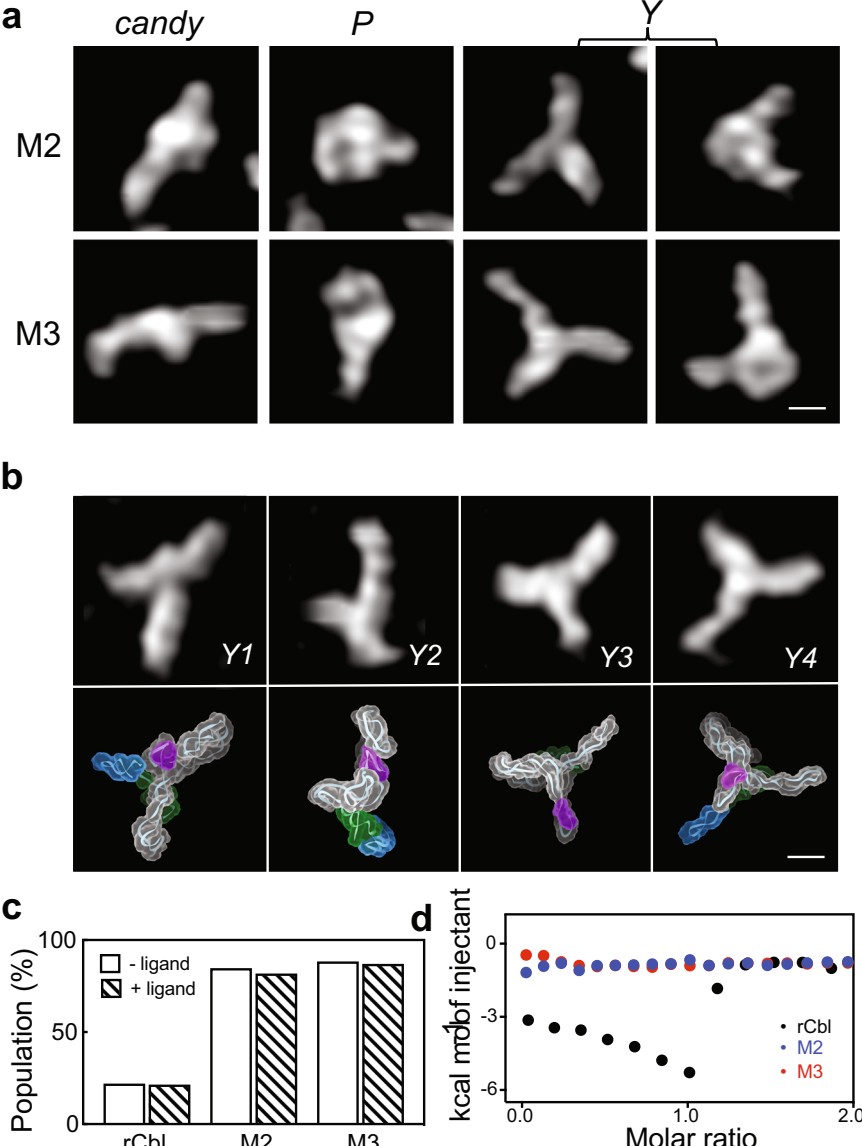

**Fig. 4 | Effects of KL interaction on rCbl folding. a** Major conformations for rCbl M2 and M3 mutants in the absence of ligand. Both mutants exhibit dominant (>80%) *Y*-shaped particles, and minor (<10% each) *P*- and *candy*-shaped particles reminiscent of the non-mutant conformations. **b** AFM images (top) and recapitulated structures (bottom) for *Y1* and *Y2* conformers of rCbl, and *Y3* and *Y4* conformers of M2 and M3 mutants, respectively. P1, P5, and P13 helices are colored in green, purple and blue, respectively. Scale bars of 5 nm are shown in the right bottom corner for (**a**) and (**b**). **c** Population tallies of *Y* conformers in the rCbl, M2 and M3 samples in the absence or presence of ligand. In all cases, the *Y* population is

unaffected by ligand binding. **d** Isotherms for rCbl (black), M2 (blue), and M3 (red). The isotherms shown are from experiments performed using freshly prepared samples at 50 µM RNA and 500 µM ligand concentrations and under the same buffer conditions ("Methods"). The raw isotherms for rCbl, M2, and M3 are shown in Supplementary Fig. 4b, c, and d, respectively. Titrations with rCbl and M3 were performed twice. The titration with M2 was performed only once since the deletion of P13 is a more extreme approach to abolishing the KL interaction, whose effect is sufficiently demonstrated in M3, which has only disrupting mutations in L13. Source data are provided as a Source Data file.

the individual molecule level. Moreover, solution AFM is amenable to large RNAs, which are more likely to adopt broader conformational space, and are often too dynamic for structure determination by X-ray crystallography or cryo-EM. Thus, we envision the use of AFM, in concert with high-resolution structural methods and computational simulation, as a new approach to analyze the conformational ensemble of a structured RNA beyond snapshots that are recorded under non-physiological conditions, and to provide a more realistic view of the solution conformational states relevant to biological function.

## Methods
### Template and primer sequences
rCbl DNA template[26]:

> TCTGATTCAGGAATTCCATAATACGACTCACTATAGGTTAAAG
> CCTTATGGTCGCTACCATTGCACTCCGGTAGCGTTAAAAGGGAAGA
> CGGGTGAGAATCCCGCGCAGCCCCCGCTACTGTGAGGGAGGACGAA
> GCCCTAGTAAGCCACTGCCGAAAGGTGGGAAGGCAGGGTGGAGGAT
> GAGTCCCGAGCCAGGAGACCTGCCATAAGGTTTTAGAAGTTCGCCTT
> CGGGGGGGAAGGTGAACA

> Forward primer: 5'-TCTGATTCAGGAATTCCATAATACGACTCACT
> ATAG-3'

> Reverse primer: 5'-TmGTTCACCTTCCCCCCCGAAGGCGAAC-3'
> rCbl M2 DNA template:

> TCTGATTCAGGAATTCCATAATACGACTCACTATAGGUUAAAGC
> CUUAUGGUCGCUACCAUUGCACUCCGGUAGCGUUAAAAGGGAAGAC
> GGGUGAGAAUCCCGCGCAGCCCCCGCUACUGUGAGGGAGGACGAAG

CCCUAGUAAGCCACUGCCGAAAGGUGGGAAGGCAGGGUGGAGGAUG
AGUCCCGAGCCAGGAGACCUGCCAUAAGGUUUUUAGAA

Forward primer: 5′-TCTGATTCAGGAATTCCATAATACGACTCACT
ATAG-3′

Reverse primer: 5′-TTmCTAAAACCTTATGGCAGGTCTCCTGGC-3′
rCbl M3 DNA template:

TCTGATTCAGGAATTCCATAATACGACTCACTATAGGUUAAAGC
CUUAUGGUCGCUACCAUUGCACUCCGGUAGCGUUAAAAGGGAAGA
CGGGUGAGAAUCCCGCGCAGCCCCCGCUACUGUGAGGGAGGACGA
AGCCCUAGUAAGCCACUGCCGAAAGGUGGGAAGGCAGGGUGGAGG
AUGAGUCCCGAGCCAGGAGACCUGCCAUAAGGUUUUUAGAAGUUCG
CCUUCGAAAGAAGGUGAACA

Forward primer: 5′-TCTGATTCAGGAATTCCATAATACGACTCAC
TATAG-3′

Reverse primer: 5′-TmGTTCACCTTCTTTCGAAGGCGAACTTCTA
AAACCTTATGGCAGGTCTC-3′

aCbl DNA template[25]:

GAAATTAATACGACTCACTATAGGTCAAATAGGTGCCGGTCCGT
GAACAACAGCCGGCTTAAAAGGGAAACCGGTAAAAGCCGGTGCGGT
CCCGCCACTGTAATTGGCCAAGCGCCAAGAGCCAGGATACCTGCCTG
TTTGATCAGCACGAATTCTGCGAGGACAGATGA

Forward primer: 5′-GAAATTAATACGACTCACTATAGG-3′
Reverse primer: 5′-mUmCATCTGTCCTCGCAGAATTCGTG-3′.

## RNA sample preparation

DNA templates were generated by PCR from synthesized linear DNA and primers (IDT, Coralville, Iowa). rCbl and mutant RNA samples were generated by in vitro T7 RNA polymerase transcription. Supplementary Fig. 1 illustrates the extensive investigation of the various factors that affect the populations of heterogeneous conformations of rCbl RNA samples: salt concentration, temperature, storage conditions, presence/absence of ligand, and RNA concentration. Of note, the sample incubated overnight at room temperature under high-salt buffer conditions and at higher RNA concentration folds into almost homogeneous dimers (Supplementary Fig. 1d). For this study, we selected the protocol which produces the sample with the highest population of binding-competent conformers based on the ITC data and AFM images, as described in the following. The transcript was purified by 6% polyacrylamide gel electrophoresis (PAGE) under native conditions. The RNA was eluted from the gel using RNA elution buffer containing 50 mM sodium acetate, pH 5.3, 2 mM EDTA, followed by buffer exchange to high-salt buffer (HSB) (10 mM HEPES, pH 7.5, 100 mM KCl, 0.1 mM EDTA, 10 mM $MgCl_2$). The size and purity of RNA was confirmed by denaturing PAGE and mass spectrometry (Agilent 6250 Accurate−Mass Q-TOF LC/MS) (Supplementary Fig. 10). For aCbl, we prepared the sample by following the exact procedure described previously[25]. The aCbl was transcribed in the presence of 0.5 mM AdoCbl, followed by 10% TBE gel purification. The RNA was eluted in 50 mM potassium acetate, pH 7.0, 200 mM KCl, then buffer-exchanged to 20 mM Tris-HCl, pH 8.0, 50 mM KCl, 10 mM $MgCl_2$.

## AFM experiments

All AFM experiments were performed under solution conditions using a Cypher VRS AFM (Asylum Research, Oxford Instrument) at 4 °C with amplitude-modulated AC mode (commonly known as tapping mode). To immobilize RNA, mica supports were treated with 1-(3-aminopropyl) silatrane (APS) (synthesized in-house) (Supplementary Discussion). 50 mM APS stock was diluted 300-fold in water just before use and coated on freshly cleaved muscovite mica (Grade V1) (Ted Pella Redding, CA). After 30 min, the mica surface was rinsed with purified water (Pico Pure Water system, Avidity, UK), and dried gently with filtered nitrogen gas. For the sample in the absence of ligand, 10 μl 20 nM rCbl was deposited on APS-mica for

~20 min and washed with 200 μl low-salt buffer (LSB). For the sample in the presence of ligand, 50 nM rCbl RNA was mixed with 1 mM AdoCbl (coenzyme B12, C0884, Sigma-Aldrich, USA) ligand first, then 10 μl mixed sample was put on APS-mica for ~15 min and washed with 200 μl ligand buffer (50 mM MES, pH 6.0, 10 mM KCl, 1 mM $MgCl_2$, 1 mM AdoCbl). To get high-resolution images, FASTSCAN-D-SS probes (Bruker) were used, which had a nominal tip radius of 1 nm, a resonance frequency of 80-140 kHz and a normal spring constant of 0.25 N/m. A pulsed blue laser (BlueDrive) was used for photothermal excitation, positioned at the base of the cantilever, while a superluminescent diode (SLD) was positioned near the head of the cantilever to detect cantilever deflection. Images were collected with a scan size of $500 \times 500$ nm$^2$, $1024 \times 1024$ pixels$^2$ and a scan rate of 1.0 Hz, with a typical initial set point of 450 mV and a free amplitude of 500 mV. The setpoint was changed after the tip approach and adjusted during the imaging based on the image quality. In total, 11 and 9 high-quality images were collected for rCbl in the absence or presence of ligand, respectively.

## AFM image processing and analysis

For images used for 3D topological structure calculation, the raw images were first processed with the following built-in settings in SPIP (Scanning Probe Image Processor) software (Metrology): plane correction by applying 3$^{rd}$-order polynomial leveling to the particle-free region, filtering by de-spiking to remove line artifacts, and Fast Fourier Transform (FFT) analysis to remove high-frequency noise. The final image resolution was increased to $4096 \times 4096$ pixels$^2$ by doubling the number of pixels twice. Single-particle images were cropped from the processed images and converted to pseudoAFM images (*.txt) with 5-Å/pixel resolution in MountainsSPIP (Digital Surf) for structure calculation. Tally analysis was done using the particle analysis function implemented in SPIP by manually selecting the particles for each class based on the topological surface. Particles with multimers or with linear nucleic acids were not counted.

## 3D topological structure calculations and validation

The following is a brief description of the method. A survey of the RNA structure database indicates that A-form duplexes are highly conserved and that A-form-like duplexes are by far the predominant building blocks in folded RNA structures[27]. The characteristic width and pitch of an RNA duplex are ~25 and 30 Å, respectively, which are on a scale similar to a sharp AFM probe and sensitive to AFM detection. Thus, an imaging resolution of 10-15 Å is routinely achievable[11].

RNA structure is largely dictated by hierarchical folding. For a given primary sequence, the closest neighboring interactions generate secondary structural elements, including duplexes, hairpins, junctions, and other types of 3D motifs, followed by forming larger domains and global folds stabilized by long-range tertiary interactions[28, 29]. Since secondary structural information is generally available prior, 3D structures of folded RNA can be recapitulated from an AFM molecular surface using dynamic fitting, an approach similar to that used in X-ray crystallography and cryo-EM[30]. Only in the case of folded RNA is this approach particularly feasible, given the AFM resolution limit and characteristics of RNA hierarchical folding. The scoring functions that underpin the dynamic fitting to AFM images take both the Cross-Correlation (CC) between the AFM molecular surface and a structure (Eqs. (1) and (2)), and energies (Eq. (3)) into consideration. The relative weight between the pseudopotential $V^{AFM}(xyz)$[31] and energy $\boldsymbol{E}$ is empirically determined by scaling factors $\theta^c$ and $\theta^{nc}$ for covalent and noncovalent interactions, respectively (Eq. (3)). To accelerate the computational speed, the dynamic fitting is computed using coarse-grained models, which are then converted into all-atom structure coordinates using RS3D[32] and

subjected to all-atom refinement with Xplor-NIH[33].

$$C^{AFM}(xyz) = \frac{\sum_i I_i^{\exp} I_i^{sim}(xyz)}{\sqrt{\sum_i \left(I_i^{\exp}\right)^2} \sqrt{\sum_i \left(I_i^{sim}(xyz)\right)^2}} \quad (1)$$

$$V^{AFM}(xyz) = \theta^{AFM} N k_B T [1 - C^{AFM}(xyz)] \quad (2)$$

Where $C^{AFM}(xyz)$, $I_i^{\exp}$, and $I_i^{sim}(xyz)$ are the correlation, experimental and simulated intensities, respectively, of the $i^{th}$ pixel at the xyz position in the molecule; $\theta^{AFM}$ is the scaling parameter of the AFM force; $N$ and $k_B$ are the total number of beads of the molecule and Boltzmann constant, respectively.

$$\begin{aligned}
E^{total} &= V^{AFM}(xyz) + \theta^c \sum E^c + \theta^{nc} \sum E^{nc} \\
&= V^{AFM}(xyz) + \theta^c \left( \sum_j E_j^{angle} + \sum_k E_k^{length} + \sum_l E_l^{dih} \right) \quad (3) \\
&+ \left( \theta^{stacking} \sum_m E_m^{stacking} + \theta^{pairing} \sum_n E_n^{pairing} + \theta^{contact} \sum_o E_o^{contact} \right)
\end{aligned}$$

where total energy $E^{total}$ is the sum of the AFM pseudopotential, covalent, and noncovalent energies for the whole molecule. The covalent energy includes bond lengths, angles, and dihedrals, whereas the noncovalent energy term includes stacking, basepairing, short- and long-range interactions, specifically van der Waals and electrostatic interactions. $\theta^{AFM}$, $\theta^c$, $\theta^{stacking}$, $\theta^{pairing}$, and $\theta^{contact}$ are the scaling factors for AFM, covalent, secondary structural interactions (stacking and basepairing), and contacts, respectively.

Since no biomacromolecular structure has ever been determined using the information truly from a single molecule, no direct evaluation of the accuracy of the recapitulated structures from AFM images could be made. Thus, we performed extensive validation using simulated data of the 268-nt catalytic domain of RNase P RNA (PDB: 3DHS)[34] as described briefly in the following. A structural model was generated using a coarse-grained MD calculation. The RMSD between the model and crystal structure is ~17 Å and the CC score between the simulated AFM molecular surface of the model and the crystal structure is between 0.4 and 0.5, depending on the percentage of noise added. The large RMSD relative to the known crystal structure and the low CC score indicate the two structures' dissimilarity, which is required to evaluate the robustness of the method. Gaussian noise with 2 σ was added to the simulated AFM molecular surfaces. A total of twelve noise levels, ±5, ±10, ±15, ±20, ±30, ±40, ±50, ±60, ±70, ±80, ±90, and ±100% of the highest value in the Z-axis were simulated and randomly added to the AFM images. The crystal structure was used as the initial structure to dynamically fit the simulated AFM images. In the case of rCbl recapitulated structures, the starting model was the crystal structure (PDB: 4GMA) with missing residues added.

The final recapitulated structure is identified and determined based on the lowest total energy and the highest CC scores (Supplementary Table 3). The RMSD between a recapitulated structure and the ground-truth structure is estimated to be ~5 Å at 40% noise level, where the CC score between the simulated noisy AFM image and the recapitulated structure from the image is greater than 0.99. The CC score drops significantly and the RMSD increases as the added noise is >40%. In practice, after image processing, the image noise is usually under 10%. A CC score between 0.98 and 0.99 is necessary to determine a recapitulated structure with RMSD of ~6 Å relative to the ground-truth structure underneath an AFM molecular surface.

## Isothermal titration calorimetry (ITC) and data analysis

ITC experiments were performed using a MicroCal iTC200 (Malvern, UK) at 25 °C, following a standard procedure of RNA-ligand titration as described previously[35] (Supplementary Fig. 4). RNA sample was buffer exchanged extensively to ITC buffer containing 50 mM MES, pH 6.0, 10 mM KCl, 1 mM MgCl₂ and AdoCbl ligand was dissolved in ITC buffer. 70 μM RNA was titrated with 700 μM AdoCbl. The blank experiment was performed by replacing RNA sample with ITC buffer and was subtracted from the experimental data during data analysis. The baseline correction and integration of the thermogram were processed using NITPIC[36]. The raw thermogram was deconvoluted into two principal components by singular value decomposition analysis (SVD) using a MATLAB script (see Supplementary Discussion). The thermodynamic parameters for each component were determined by nonlinear regression fitting using either a one-site or two-site binding model.

The raw thermogram of AdoCbl binding to rCbl appeared unconventional, and differed from a typical exothermic binding heat compensation followed by an endothermic response (Supplementary Fig. 11a), suggesting the AdoCbl binding triggers a substantial conformational change in the riboswitch aptamer via an induced-fit binding scenario, which is further evidenced by our AFM tally results (main text) in the presence and absence of AdoCbl.

To determine the number of states associated with the AdoCbl binding reaction and rationally reconstruct the ITC thermogram with all significant components (Supplementary Fig. 11b–d), a series of titration peaks of the ITC thermogram (Supplementary Fig. 11a) were subjected to SVD analysis using MATLAB (MATLAB and Statistics Toolbox Release 2021a, The MathWorks, Inc., Natick, Massachusetts, United States). The heat compensation profiles related to each titration point were used to produce an $m \times n$ matrix, M, and used as input for SVD analysis, where $m$ corresponds to the number of the recorded heat compensation profiles and $n$ corresponds to the number of titration points at different ligand-to-RNA ratios. To unbiasedly determine the number of significant components (Supplementary Fig. 11e), the normalized autocorrelation coefficients of individual singular values were calculated with a minimum threshold of 0.75 as a selection filter (dashed line in Supplementary Fig. 11f). Those SVD components with autocorrelation coefficients less than 0.75 were considered systematic noise and were not used to reconstruct the ITC thermogram and to derive the related thermodynamic parameters. The isotherms integrated from selected thermograms were subjected to nonlinear regression fitting by the 1:1 one-site binding model or two-site independent binding model, as described previously[37], and fitting results referring to each SVD component were tabulated (Supplementary Table 2). The dissociation constants, binding stoichiometry, and thermodynamic parameters of AdoCbl binding to rCbl were derived by fitting the deconvoluted isotherms to the numerical solution of the explicit ordinary differential equation of the Wiseman isotherm formulation[37, 38] using MATLAB and GraphPad Prism version 9.4.1 (GraphPad Software, La Jolla, California, USA). The choice of model, i.e., one-site or two-site binding model, was determined using the F-test.

To confirm that the mutant $Y$ conformers were inactive, the same ITC experiments were conducted for both rCbl and mutants (M2 or M3) under the same solution conditions and experimental settings, except for RNA and ligand concentrations, which were 50 μM RNA and 500 μM, respectively.

## Small-angle X-ray scattering (SAXS) experiments and data analysis

The SAXS experiments were carried out at the 12-ID-B beamline of the Advanced Photon Source (APS), Argonne National Laboratory. Photon energy was 13.3 keV ($\lambda = 0.932$ Å). A sample-to-detector distance of 1.9 m was used to obtain a $q$ range of $0.005 < q < 0.88$ Å$^{-1}$. Here, $q$ is the magnitude of the scattering vector, $q = (4\pi/\lambda)\sin\theta$, where $2\theta$ is the scattering angle and $\lambda$ is the wavelength of the

radiation. Ideally, the concentration of rCbl for SAXS should be the same as for the AFM experiments. However, the concentration of 20 nM for AFM is too low to produce reasonable SAXS signals. To select a concentration that is as low as possible and yet produces reasonable scattering curves after background subtraction, a wide range of RNA concentrations from 25 nM to 2 μM were analyzed. For RNA samples complexed with ligand, the concentrations of rCbl were in the same range of 25 nM to 2 μM, with the molar ratio of rCbl to the ligand of 1:10. Buffers containing ligand at concentrations in the range of 225 nM to 18 μM were measured accordingly, and used for background subtraction for ligand-containing samples. To minimize radiation damage and obtain a good signal-to-noise ratio, 225 images were taken for each sample and buffer using a flow cell with an exposure time of 1 s. The two-dimensional scattering patterns were collected using a Pilatus 2 M detector and converted to one-dimensional SAXS curves through radial averaging after solid angle correction and then normalizing with the intensity of the transmitted X-ray beam. Data were then averaged after the elimination of outliers using the software package developed at beamline 12-ID-B. The information about the samples for SAXS experiments, the SAXS data collection and analysis, and the software used are listed in Supplementary Table 6 and the notes thereafter.

The radius of gyration, $Rg$, and intensity at angle zero, $I(0)$, were generated from the Guinier plot in the range of $qR_g < 1.3$. For comparison, $Rg$ and $I(0)$ were also calculated in real and reciprocal spaces using the program GNOM in the $q$ range up to $0.30\,\text{Å}^{-1}$[39]. The pair-distance distribution function $P(r)$ and maximum dimension ($D_{max}$) were also calculated using GNOM. The molecular weights were estimated based on the method of correlation volume, $Vc$, using the formula for RNA[40]. The obtained structural parameters and molecular weights are listed in Supplementary Table 4.

The SAXS data for rCbl at the concentration of 1 μM were selected for the ensemble fitting as this was the lowest concentration with a reliable scattering signal (Supplementary Fig. 3). The synthesized scattering intensity from an ensemble of conformations can be described as:

$$I_{syn}(q_i) = \sum_{k=1}^{N_{ens}} I_{calc}^k(q_i) \cdot v_k$$

Where $v_k$ and $I_{calc}^k(q)$ are the volume fraction and calculated scattering intensity from the $k^{th}$ component/conformation, respectively. $N_{ens}$ is the number of conformations in the ensemble. All conformations used for fitting were obtained by AFM. The scattering profiles for each conformation in the ensemble were calculated in the $q$ range $0 < q < 0.50\,\text{Å}^{-1}$ using Crysol 3.0[41]. The simple average of the back-calculated scattering profiles, where conformational species were equally populated, did not agree with the experimentally measured SAXS data. Instead, the volume tallies from AFM (Fig. 1e) were used to generate initial values of volume fractions $v_k$ for each conformation, which were then optimized to achieve optimal fit (Fig. 2e and Supplementary Table 5).

The goodness-of-fit of the synthesized scattering intensity from an ensemble to the SAXS experimental data was evaluated by comparing the synthesized profile, $I_{syn}(q)$, with the experimental one $I_{saxs}(q)$.

$$\chi^2 = \frac{1}{N_q - 1} \sum_{i=1}^{N_q} \left[ \frac{I_{saxs}(q_i) - a \cdot I_{syn}(q_i) - b}{\sigma(q_i)} \right]^2$$

where, $a = \sum_{i=1}^{N_q} I_{saxs}(q_i) \cdot I_{syn}(q_i) / \sum_{i=1}^{N_q} I_{saxs}(q_i) \cdot I_{saxs}(q_i)$, $b$ is background offset, $N_q$ is the number of experimental points, $\sigma(q)$ is the experimental error.

## Reporting summary

Further information on research design is available in the Nature Portfolio Reporting Summary linked to this article.

## Data availability

The SAXS data generated in this study have been deposited in the SASBDB under accession codes SASDQG7 and SASDQH7 for rCbl in the absence and presence of ligand, respectively. All back-calculated SAXS curves based on observed conformers together with the synthesized SAXS and experimental SAXS curves are available at https://home.ccr.cancer.gov/csb/pnai/data/InfoForRNAHeterogeneityStudy/SAXS_plots/. All structural model coordinates and computation files for rCbl are available at https://home.ccr.cancer.gov/csb/pnai/data/InfoForRNAHeterogeneityStudy/. PDB coordinates for rCbl, aCbl, and RNase P used for the analyses in this study are 4GMA, 6VMY, and 3DHS, respectively. Source data are provided with this paper.

## Code availability

All codes used for the study can be downloaded at https://home.ccr.cancer.gov/csb/pnai/data/InfoForRNAHeterogeneityStudy/.

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

## Acknowledgements

This work was funded by the Intramural Research Program of the National Cancer Institute, National Institutes of Health (Y.-X.W.) and in part with federal funds from the National Cancer Institute, National Institutes of Health, under contract HHSN261201500003I (R.N.). The content of this publication does not necessarily reflect the views or policies of the Department of Health and Human Services, nor does mention of trade names, commercial products, or organizations imply endorsement by the U.S. Government. We thank Will F. Heinz for technical assistance in AFM, Benjamin Miller, Steve Fellini and Susan Chacko of the NIH HPC computing facility for allocating computing and storage resources for the project, and Dr. Jeffrey N. Strathern for his vision about RNA biology.

## Author contributions

J.D. and P.Y. prepared samples; J.D. recorded all AFM images; J.D. and Y.-X.W. performed all calculations of the structures; Y.R.B. and C.D.S. wrote all computation codes; L.F. recorded and analyzed SAXS data; L.F. and Y.-X.W. interpreted experimental SAXS data; J.D., Y.-T.L., and S.G.T. recorded ITC data; Y.-T.L. analyzed/interpreted the ITC data; B.M. and R.N. participated in the initial phase of the project; J.R.S. for very careful examination of the electron density map of the rCbl crystal structure; A.R. and J.Z. for many insightful discussion; Y.-X.W. conceptualized, designed the project, drafted the manuscript; all authors contributed to the revision.

## Funding

## Competing interests

The authors declare no competing interests.
