## [Peer Review File · Nature Communications]

Visualizing RNA conformational and architectural heterogeneity in solutionEditorial Note: Parts of this Peer Review File have been redacted as indicated to maintain the confidentiality of unpublished data.

Reviewers' comments:

Reviewer #1 (Remarks to the Author):

Noteworthy Results

The manuscript by Ding et al. uses AFM and SAXS to examine the conformational states of a cobalamin riboswitch from *Thermoanaerobacter tengcongensis*. The authors argue that AFM offers the advantage of being a solution technique that is capable of interrogating the numerous conformational states adopted by the riboswitch in the absence of presence of ligand. The authors extend the usefulness of AFM by using course-grain MD and dynamic fitting to produce low-resolution all atom riboswitch conformers that describe the various monomeric, dimeric and multimeric states of the riboswitch. A main take home message is that RNA is unlikely to obey a 1 sequence 1 fold relationship, which has implications for the ability to predict folds in a manner currently employed by deep learning approaches on proteins. The work is laudable for its innovative use of AFM and the ability to use the solution information to describe conformational ensembles that describe SAXS profiles measured for the riboswitch under comparable solution conditions. Despite this enthusiasm, the reviewer has significant doubts about the approach and interpretation. Due to the choice of a thermophilic RNA, it is also unclear whether the results are widely applicable to other RNAs, which could limit the impact of the findings. Several suggestions are made to improve the manuscript.

Is the Methodology Sound?

Page 7, Heterogeneous Conformers. The authors state that the “raw isotherm and thermogram” are shown. However, it is not really “raw” unless it is completely uncorrected (i.e., not baseline adjustment, buffer subtracted etc, which is stated in the Methods). Moreover, The reviewer sees only the isotherm drawn through points generated from the integrated heats of injection. The thermogram itself is not shown. However, the actual thermograms with injection heats as a function of time must be shown for quality control reasons stated below.

Fig. 2. The injections should be shown that gave rise to the isotherms in Fig. 2e and elsewhere. This is because there is useful information in the injections that allows the reader to evaluate the quality of the experiment. For example, did the injection return to baseline (i.e., enough time between injections), were there bubbles and a properly adjusted baseline, were there heats of dilution from the ligand, were there buffer mismatches etc. As another point, it is unclear how many times each ITC experiments was repeated? The results should be replicated multiple times (at least 2x) with the same result, including the fractional Napp values.

Fig. 2e. How was singular value decomposition performed? The Methods say this was performed in MatLab using a script. Please provide the script so that the analysis can be reproduced by the community.

The reviewer notes that the RNA studied is from *T. tengcongensis*, a thermophilic organism. While the authors varied a several conditions during the folding, they did not attempt analyze the RNA at a temperature at which this organism is found. A previous study [Marszalkowski et al (2021) RNA 27, 1204-1219] found that in vitro transcribed RNase P RNAs from *Thermus thermophilus* that were incubated at temperature higher than 37 oC had improved catalytic activity. Marszalkowski et al speculated that this high temperature incubation was required because “the native folding transition has an extraordinarily high activation barrier” — likely a trait shared structured RNAs from thermophiles. While RNA from thermophiles usually make suitable candidates for crystallization, due to a high stability at room temperature, it appears to be commonplace to fold the RNA at ≥ 65 oC to ensure folding similar to the native state. Based on this, it seems unlikely that the cobalamin riboswitch from *T. tengcongensis* folds the same at 37 oC solely based on in vitro transcription (presumably) and using a non-*T. tengcongensis* polymerase. This aspect of the approach leaves serious doubts about the folding landscape, which is likely stuck in a number of local minima due to the thermophilic nature of the RNA. This point raises the question of whether the observations here are applicable to mesophilic riboswitches in general. The reviewer recommends conducting a comparable analysis on a mesophilic cobalamin riboswitch for comparison or conducting folding and native purification under conditions better suited to a thermophilic RNA.

Given the last point, the authors should consider the effect of temperature on the AFM studies. As mentioned in the previous point, this RNA is derived from a thermophilic bacterium and likely does not sample all conformations at 25 oC because it has evolved to be functional (i.e. sample these states) at ≥ 65 oC.

Fig 3. Beyond modeling, what other evidence exists that supports the KL as a motif required for heterogeneous dimer formation? In other words, given the untested nature of the authors’ approach, additional evidence seems important to support the coarse-grain models and dynamic fitting that forms the basis of this work. What independent evidence supports the modeling approach? This problem appears well suited to single molecule FRET studies or single molecule force extension analysis.

The authors state that a major advantage of the approach is that it provides direct visualization of a specific RNA sequence to provide insight into the heterogeneous conformations accessible in solution conditions that approximate the physiological state. However, on page 16 the Methods state that the authors use 1-(3-aminopropyl) silatrane (APS) to immobilize RNA samples on mica. What evidence supports the use of APS as a non-denaturing reagent that preserves the “native” structure of particles immobilized on the mica surface? Since the immobilization depends on covalent bond formation, it is conceivable that the immobilized structures must be partially unfolded. This does not seem to be a true solution state but rather one that is hydrated under controlled conditions. The authors should be more cautious in asserting that the AFM condition is physiologically relevant, especially since the data are

based on tapping of the RNA surface with a force probe.

Page 15. How confident are the authors that the transcription products are the desired riboswitch? T7 polymerase is known to form untemplated sequences caused by priming of short RNA hairpins as they are transcribed [Nucleic Acids Research, Volume 46, Issue 18, 12 October 2018, Pages 9253–9263]. This suggests that some transcription products purified by native PAGE might not actually be the desired sequences or folds. The reviewer believes that purified RNAs should be verified by sequencing to prove that the transcripts comprise the desired riboswitch sequence. Next generation sequencing is readily amenable to this problem. The reviewer also believes that there are better approaches for native purification in lieu of gel electrophoresis. For example, Kieft and Batey [RNA 2007 13(8):1384-9] described a native purification method in which the co-transcriptionally folded RNA is affinity purified by MS2 coat protein hairpins and cleaved by a glmS riboswitch. This approach assures that the transcription product has some function and the correct 3'-end sequence for affinity purification. For quality control, the authors should provide evidence that their transcripts are the correct sequence. A denaturing gel with serially diluted RNA is also important to show for quality control of AFM and SAXS experiments.

Page 20. To assess possible concentration-dependent oligomerization, the authors should show 1-D scattering profiles for various riboswitch concentrations over the same momentum transfer range. Here, it is important to show that the thermodynamic ensemble is not changing appreciably in a concentration dependent manner over the concentration range used to create the scattering profile. The reviewer expects to see that the components used from AFM are an accurate description of the solution ensemble, which is not distorted by concentration dependent aggregation.

Page 20. The Guinier plots used to calculate R_g values should be included for transparency and quality control. These plots can also provide confidence that samples are not aggregating (self-associating) or experience radiation damage, which would be apparent in the Guinier analysis.

Page 20. Why didn't the authors use the average scattering profile from multiple concentrations for ensemble fitting? What was the rationale for using only 1 μM ? This concentration is significantly higher than the AFM concentration.

Areas that require improved clarity

Page 3. Intro. What do the authors mean by "driving the molecules to uniformity under extreme conditions"?

Page 4: It is a little misleading to say, "molecules can be observed in their native states without any manipulation, e.g. labeling, freezing, staining or crystallization" when you immobilize the RNA on a surface. While it is true that you do not need to attach a bulky fluorescent label, the RNA is unnaturally tethered to a surface.

As a point of clarity, it is unknown what RNA sequence was used for this study. This point must be rectified for the reader. Is the riboswitch the same sequence as that crystallized by Johnson, J.E., Reyes, F. E., Polaski, J.T. & Batey, R.T. (2012) *Nature* 492, 133-137? If so, this construct contains mutations to aid crystallization (see Supplementary Fig. 1 of that paper). The authors should add a figure or table to the supplement to show the exact sequences transcribed for this work. Does the construct contain the expression platform? This is a transcriptional riboswitch.

Page 8, Extended Figure 5. The sequence of the riboswitch cannot be made out due to the small size and low resolution of this figure. Please make this figure clearer so that the reader can understand changes in the sequence associated with M2 and M3 in Fig. 4. A more detailed depiction of the kissing loop (e.g. a stick and ball diagram) would also be helpful to understand the molecular basis for this interaction. This should be called out early in the manuscript. It would help orient the reader to label which portion of the rCbl construct contains the expression platform in supplementary fig. 5. Also Please remove the red squiggly underline in supplementary fig. 5. The purpose is unclear.

Fig. 2. It is difficult to see the synthesized SAXS curve, stated to be a red solid line. The line should be moved in front of the black line.

Page 15. please provide the company that produces Pico pure water.

Page 16. What criterion was used to assign the AFM resolution as 5 Å? The definition of resolution should be provided for the reader since most members of the journal audience are unlikely to know. Did samples have an internal reference? Please also state why the resolution is limited by the probe tip.

Page 17. The application of 2D FFT and 3rd order polynomial flatten, as well as digital resolution extension appears to be described twice. Was this really the case? See the prior paragraph on page 16.

Evidence Supports Conclusions?

Page 17. The details of 3-D structure calculations are described in an accompanying paper that is not peer reviewed. This prevents the reader from independently assessing the approach and results. To rectify this shortcoming, the reviewer believes additional Extended information should be presented to describe the approach and the results. In particular, how was the process validated? Why not use proteins (one sequence = one structure) as a benchmark for the methodology? An additional concern, is that it seems plausible that some RNAs will be immobilized on mica to create preferred orientations that would complicate 3-D envelope restorations — like cryo-EM. How is this problem overcome? The methods should also describe the origin of the starting models used for coarse-grained MD. The accompanying methods paper suggests models were used from the PDB. Please clarify.

Extended Data 1: Why do the AFM and SAXS populations show such large differences in the population contributions of various species? The Y conformation is especially different. This is worth commenting upon since the AFM particles were used to model the SAXS scattering profile.

Extended Table 1. What is the basis for the values shown in this table?

Is it from coarse-grained dynamic fitting? Please state this in the table title and provide some explanation of the parameters and their origins.

Rg values in Extended Data Table 3 are inconsistent in terms of significant figures. Also, a precision of 0.1 Å seems more reasonable than 0.01 Å.

Impact of Work/Significance to Field

The Discussion ends somewhat abruptly. It would be helpful to relate the findings to other riboswitches or RNA systems with known heterogeneity problems. For example, the Bsu yvrC cobalamin riboswitch (Chan & Mondrgaon NAR 2020) proved extremely sensitive to folding conditions and required co-transcriptional folding followed by native gel electrophoresis. The work also suggested that the different conformations of the Bsu riboswitch change in response to ligands (e.g., adenosyl cobalamin versus hydroxy or methyl cobalamin). Is it possible that the conformations observed here are responsive to binding other ligands? Did the authors try other ligands besides adenosyl cobalamin to see if they alter the observed solution conformation? Are there other examples of large folded RNAs with rugged folding landscapes that would be amenable to the type of analysis shown here? For example, group I introns or group II introns?

Riboswitches are located within long mRNA transcripts and are widely accepted to act in cis vs in trans. A major finding from this work is that ligand binding induces dimerization with differing modes of contact. However, it is unclear how this finding relates to riboswitch function — which is a driving factor of most structural studies. As the authors know, riboswitches that regulate transcription termination have a narrow temporal window during co-transcriptional folding to affect transcription [Watters, K.E. et al. (2017) Nat. Struct. Mol. Biol. 12, 1124-1131]. Thus, it is unlikely that transcription regulating riboswitches have time (or another RNA with which) to dimerize. While translation regulating riboswitches have a longer window to “switch”, it is unlikely that two mRNAs containing the same riboswitch would find one another inside the packed cell environment to dimerize. Moreover, it has been recently shown that small riboswitches can regulate translation co-transcriptionally [Chatterjee S., Chauvier, A., Dandpat, S.S., Artsimovitch, I. & Walter, N.G. (2020) Proc. Nat. Acad. Sci. U. S. A. 118 e2023426118], wherein the riboswitch is sterically inhibited from dimerizing due to the transcription and translation machinery. The authors are correct that crystal structures of RNAs don't capture flexibility observed in solution experiments and the AFM presented shows distinct oligomers of this RNA in the presence of ligand. However, given previous work, it is unclear how the findings of the current manuscript relate to previously established riboswitch functions in the context of an mRNA. It is also not discussed whether the large changes observed here are representative of all riboswitches, just this riboswitch, or whether the conditions of the AFM experiment exaggerate motions — or represent motions— that may be present inside the crowded folding environment of the cell. These are comments that should be addressed in the authors' rather brief Discussion or a Supplemental Discussion that addresses the larger relevance of the work.

Reviewer #2 (Remarks to the Author):

The paper 'Visualizing RNA conformational and architectural heterogeneity in solution' by Jienyu Ding, Yun-Tzai Lee, Yuba Bhandari, Lixin Fan, Charles D. Schwieters, Ping Yu¹, Sergey G. Tarosov, Jason R. Stagno, Buyong Ma, Ruth Nussinov, Alan Rein, Jinwei Zhang, Yun-Xing Wang reports about an AFM imaging study of a single RNA sequence and shows that RNAs are conformationally heterogeneous structures.

Timeliness and importance: X-ray crystallography and, most recently and powerfully, cryo-EM are used to solve structures of biomolecules rather routinely. However, these techniques are not well adapted to study RNA due to their small size and flexibility. Thus, researchers often relied on theoretical RNA structures. Breaking with current concepts, this study shows that RNAs under physiologically relevant conditions are much more diverse than thought. Given the general importance and current emerging interest in RNA molecules, this study is very timely and important.

For the sake of clarity and as a reference system, the authors chose to study RNA cobalamin riboswitches that can equally be studied in crystals and using other methods.

The AFM single molecule data is of excellent quality for which this reviewer congratulates the authors.

The comparison with ensemble SAXS curves is elegant and convincing, and the complementary investigation of binding using ITC gives us confidence in the functional relevance of the findings.

One conformer is evidenced to be inactive and good reason is presented as to why this could be the case.

The paper is clearly and concisely written. I have only one criticism. Despite AFM being the main enabling method in this work, there is literally a single AFM reference (ref 10), which is in my view not even the best chosen. The AFM field has seen a series of improvements in the last decade for the improvement of speed and resolution that merit mention as this paper convincingly proposes AFM as a powerful technique for RNA imaging.

Reviewer #3 (Remarks to the Author):

The manuscript of Ding et al. presents AFM structures of a cobalamin riboswitch. The main finding of the work is that the riboswitch does not adopt a single conformation, but a whole ensemble of them. Based on this, the authors conclude that RNA is much more structurally heterogeneous than previously

assumed and that there is no correspondence of one structure per sequence. There is a serious flaw with the assumptions that undermines all the conclusions and makes the manuscript unsuitable for publication. As it is, this work should not be published. The problems described below have been discussed recently in a review by Lennon and Batey (JMB, 2022) on cobalamin riboswitches. The authors should consult this work as it relates to their problems.

It is well known that RNA produced by *in vitro* transcription followed by purification is prone to be misfolded. There are many possible reasons for this, but a likely explanation is that polymerases commonly used to produce RNA, such as T7 RNA polymerase, are very fast and may pause during transcription. These pauses during *in vitro* transcription are not the same as the ones that occur using the polymerase that matches the RNA being produced. RNA molecules fold co-transcriptionally, which means that polymerase pauses are crucial to allow some folding events to occur. *In vitro* RNA folding protocols are also prone to produce misfolded structures unless extreme care is taken and it is not always possible to find a suitable folding protocol. In the case described in the manuscript, the riboswitch is from a thermophilic organism and *in vitro* transcription was used for RNA production (the polymerase used is not mentioned, but it would typically be T7 RNA polymerase). It is highly likely that the RNA produced is heterogeneously folded due to the production process. Furthermore, many transcriptional riboswitches bind their ligand during transcription and folding of the RNA. Once transcribed, it may not be possible to bind the ligand. As the authors noticed, some of the interactions that may not be present are tertiary interactions, such as kissing loop interaction, which may not be properly formed during transcription if the RNA folds into a conformation that prevents the tertiary interactions to occur. The same RNA was used for crystallographic studies, but in this case the crystallization process itself may serve as a way to select properly folded molecules.

It is not surprising that all the experiments are consistent as they all used the same RNA produced in the same manner. It is also not surprising that only one conformation is cobalamin-binding capable, arguing for only one correctly folded state. In the end, it is always important to remember that *in vitro* observations have to recapitulate *in vivo* results. In the case of riboswitches, the observation is that in the cell these RNA elements are efficient and work well as on/off switches. There is not a graduated response where only a proportion of them work. They all work. This means that *in vivo* they are all correctly folded and capable of binding their ligand. This is not what the authors observed using their likely misfolded molecules.

Reviewers' comments:

Reviewer #1 (Remarks to the Author):

Noteworthy Results

The manuscript by Ding et al. uses AFM and SAXS to examine the conformational states of a cobalamin riboswitch from *Thermoanaerobacter tengcongensis*. The authors argue that AFM offers the advantage of being a solution technique that is capable of interrogating the numerous conformational states adopted by the riboswitch in the absence of presence of ligand. The authors extend the usefulness of AFM by using course-grain MD and dynamic fitting to produce low-resolution all atom riboswitch conformers that describe the various monomeric, dimeric and multimeric states of the riboswitch. A main take home message is that RNA is unlikely to obey a 1 sequence 1 fold relationship, which has implications for the ability to predict folds in a manner currently employed by deep learning approaches on proteins. The work is laudable for its innovative use of AFM and the ability to use the solution information to describe conformational ensembles that describe SAXS profiles measured for the riboswitch under comparable solution conditions. Despite this enthusiasm, the reviewer has significant doubts about the approach and interpretation. Due to the choice of a thermophilic RNA, it is also unclear whether the results are widely applicable to other RNAs, which could limit the impact of the findings. Several suggestions are made to improve the manuscript.

We thank the reviewer for appreciating the novelty and value of this study. The reviewer made several constructive suggestions, which we greatly appreciate, and we have incorporated changes to the manuscript where appropriate. The reviewer's primary concern seems to be centered around the question of whether the results from this study conducted at low temperature using a thermophilic RNA could be relevant to the conformational landscape at higher temperatures. As this concern is mentioned in the comments below, we address this concern in our responses to those particular comments.

Is the Methodology Sound?

Page 7, Heterogeneous Conformers. The authors state that the "raw isotherm and thermogram" are shown. However, it is not really "raw" unless it is completely uncorrected (i.e., not baseline adjustment, buffer subtracted etc, which is stated in the Methods). Moreover, The reviewer sees only the isotherm drawn through points generated from the integrated heats of injection. The thermogram itself is not shown. However, the actual thermograms with injection heats as a function of time must be shown for quality control reasons stated below.

We thank the reviewer for the comment and suggestion. We have added the raw isotherms and thermograms in the Supplementary Figure 4 (page 14 in SI).

Fig. 2. The injections should be shown that gave rise to the isotherms in Fig. 2e and elsewhere. This is because there is useful information in the injections that allows the reader to evaluate the quality of the experiment. For example, did the injection return to baseline (i.e., enough time between injections), were there bubbles and a properly adjusted baseline, were there heats of dilution from the ligand, were there buffer mismatches etc. As another point, it is unclear how many times each ITC experiments was repeated? The results should be replicated multiple times (at least 2x) with the same result, including the fractional Napp values.

In the revision, we included the thermogram showing heats of injection in Fig. 2 and additional ITC data in the supplementary information (Supplementary Figure 4, page 14 in SI) to demonstrate the quality of the data and their interpretation. This includes the thermogram for ligand injection into the buffer alone, which was used for baseline subtraction. The experiments were performed a number of times under various conditions.

Fig. 2e. How was singular value decomposition performed? The Methods say this was performed in MatLab using a script. Please provide the script so that the analysis can be reproduced by the community.

The MatLab script has been added to the supplementary information (pages 3-8 in SI).

The reviewer notes that the RNA studied is from *T. tengcongenensis*, a thermophilic organism. While the authors varied a several conditions during the folding, they did not attempt analyze the RNA at a temperature at which this organism is found. A previous study [Marszalkowski et al (2021) RNA 27, 1204-1219] found that in vitro transcribed RNase P RNAs from *Thermus thermophilus* that were incubated at temperature higher than 37 oC had improved catalytic activity. Marszalkowski et al speculated that this high temperature incubation was required because “the native folding transition has an extraordinarily high activation barrier” — likely a trait shared structured RNAs from thermophiles. While RNA from thermophiles usually make suitable candidates for crystallization, due to a high stability at room temperature, it appears to be commonplace to fold the RNA at ≥ 65 oC to ensure folding similar to the native state. Based on this, it seems unlikely that the cobalamin riboswitch from *T. tencongenensis* folds the same at 37 oC solely based on in vitro transcription (presumably) and using a non-*T. tencongenensis* polymerase. This aspect of the approach leaves serious doubts about the folding landscape, which is likely stuck in a number of local minima due to the thermophilic nature of the RNA. This point raises the question of whether the observations here are applicable to mesophilic riboswitches in general. The reviewer recommends conducting a comparable analysis on a mesophilic cobalamin riboswitch for comparison or conducting folding and native purification under conditions better suited to a thermophilic RNA.

Given the last point, the authors should consider the effect of temperature on the AFM studies. As mentioned in the previous point, this RNA is derived from a thermophilic bacterium and likely does not sample all conformations at 25 oC because it has evolved to be functional (i.e. sample these states) at ≥ 65 oC.

The reviewer is correct in pointing out that the RNA used in this study is from a thermophile, and was transcribed with T7 RNA polymerase at 37 °C, purified under non-denaturing conditions, and imaged at 4 °C to maximize image quality and resolution. The reviewer’s point that the folding landscape of this riboswitch under these experimental conditions may in fact be different from that under thermophilic and/or in vivo conditions is also a valid one. Nevertheless, we would like to point out that this study does not aim to delineate (necessarily) a complete ensemble of actual atomic structures associated with the riboswitch under its native cellular conditions, but rather to demonstrate that the single sequence of this RNA is able to fold into heterogeneous conformations, all of which, except for one, are active in terms of ligand binding capability, and thus biologically relevant. From a purely energetic standpoint, we would contend that these structures depict in part the aptamer’s folding landscape and heterogeneous conformations that might be present along the folding trajectory. Again, we emphasize that only one of these conformers (Y-shaped) is energetically trapped (misfolded), as our ITC data show that ~80% of the conformers that exist in solution are capable of ligand binding, and that this process involves conformational changes. It is possible that more conformations exist at elevated temperatures, but this

would not alter the intended purpose of this manuscript nor its stated conclusion: “one sequence can fold into multiple (active) conformations.” Given this fact, we do not think that additional experiments at higher temperatures are warranted, nor would they be very practical to conduct on RNA samples at the resolution sufficient for detailed structure analysis, in which high temperature is likely to have negative effects on particle immobilization, imaging quality due to increased motion, and probe stability. To our knowledge, no such experiments have been attempted for high resolution imaging on biological molecules in solution.

To address the reviewer’s suggestion of using a mesophilic cobalamin riboswitch, we performed the same experiment using cobalamin riboswitch from *B. subtilis*, a mesophilic strain. As we expected, we observed almost identical behavior of heterogeneous aptamer conformations as seen with that from *T. tengcongensis*. The *B. subtilis* results reaffirm that heterogeneity is not unique to that of *T. tengcongensis* or sample preparation protocols. We provided this new result in the supporting info of the revised manuscript (Supplementary Fig. 2, page 12 in SI).

Thus, whether the cobalamin riboswitch is from thermophile or mesophile, the conformational heterogeneity at 1 mM Mg^{2+} is similar, and not by coincidence or an artifact of thermophilic RNA transcribed at lower temperatures. Moreover, the AFM observations are conducted in solution at 1 mM Mg^{2+} , which is more relevant than snapshot crystal structures obtained under harsh chemical conditions (300 mM Mg^{2+}).

Fig 3. Beyond modeling, what other evidence exists that supports the KL as a motif required for heterogeneous dimer formation? In other words, given the untested nature of the authors’ approach, additional evidence seems important to support the coarse-grain models and dynamic fitting that forms the basis of this work. What independent evidence supports the modeling approach? This problem appears well suited to single molecule FRET studies or single molecule force extension analysis.

As discussed in the manuscript, we demonstrate by AFM without modeling (Fig. 4a-b, page 17 in main text; Supplementary Fig. 6, page 16 in SI), and the ITC results (Fig. 4d, page 17 in main text; Supplementary Fig. 4c-d, page 14 in SI) the significance of the KL interaction in the proper folding of the aptamer as well as dimer formation. The mutation data show that the inability to form the KL interaction results in ~80% misfolded (Y-conformation) and no indication of dimers, strongly suggesting that dimerization must occur through KL interactions, consistent with our structure analysis. We feel that the complete illustration and validation of the methodology are outside the scope of this manuscript, and will be presented in detail in a separate manuscript to be published elsewhere. smFRET or single molecule force extension experiments are powerful tools, but they are not direct methods and require calibration/verification using structural models. Such experiments are worthy of separate independent studies, but are thus beyond the scope of the current study.

The authors state that a major advantage of the approach is that it provides direct visualization of a specific RNA sequence to provide insight into the heterogeneous conformations accessible in solution conditions that approximate the physiological state. However, on page 16 the Methods state that the authors use 1-(3-aminopropyl) silatrane (APS) to immobilize RNA samples on mica. What evidence supports the use of APS as a non-denaturing reagent that preserves the “native” structure of particles immobilized on the mica surface? Since the immobilization depends on covalent bond formation, it is conceivable that the immobilized structures must be partially unfolded. This does not seem to be a true solution state but rather one that is hydrated under controlled conditions. The authors should be more

cautious in asserting that the AFM condition is physiologically relevant, especially since the data are based on tapping of the RNA surface with a force probe.

The reviewer's statement of covalent immobilization is factually incorrect. The immobilization is not via covalent alteration of the RNA structure. The functionalization of mica with APS is widely used for the immobilization of nucleic acids primarily through the electrostatic interactions between protonated amino groups of the APS-mica substrate and the negatively charged nucleic acid backbone. The low charge density of APS-mica allows reliable imaging of nucleic acids, protein-nucleic acid complexes and other biological samples, which has been demonstrated in many papers¹⁻³.

1. Shlyakhtenko, L.S. et al. Silatrane-based surface chemistry for immobilization of DNA, protein-DNA complexes and other biological materials. *Ultramicroscopy* **97**, 279-87 (2003).
2. Lyubchenko, Y.L., Shlyakhtenko, L.S. & Gall, A.A. Atomic force microscopy imaging and probing of DNA, proteins, and protein DNA complexes: silatrane surface chemistry. *Methods Mol Biol* **543**, 337-51 (2009).
3. Stumme-Diers, M.P., Stormberg, T., Sun, Z. & Lyubchenko, Y.L. Probing The Structure And Dynamics Of Nucleosomes Using Atomic Force Microscopy Imaging. *J Vis Exp* (2019).

Page 15. How confident are the authors that the transcription products are the desired riboswitch? T7 polymerase is known to form untemplated sequences caused by priming of short RNA hairpins as they are transcribed [Nucleic Acids Research, Volume 46, Issue 18, 12 October 2018, Pages 9253–9263]. This suggests that some transcription products purified by native PAGE might not actually be the desired sequences or folds. The reviewer believes that purified RNAs should be verified by sequencing to prove that the transcripts comprise the desired riboswitch sequence. Next generation sequencing is readily amenable to this problem. The reviewer also believes that there are better approaches for native purification in lieu of gel electrophoresis. For example, Kieft and Batey [RNA 2007 13(8):1384-9] described a native purification method in which the co-transcriptionally folded RNA is affinity purified by MS2 coat protein hairpins and cleaved by a glmS riboswitch. This approach assures that the transcription product has some function and the correct 3'-end sequence for affinity purification. For quality control, the authors should provide evidence that their transcripts are the correct sequence. A denaturing gel with serially diluted RNA is also important to show for quality control of AFM and SAXS experiments.

We have added mass spectrometry data and denaturing-PAGE results in the supporting information to show that the purified RNA represents a correct, homogeneous sequence (Supplementary Fig. 10, page 20 in SI). In regards to the purification method for obtaining functional species, the RNA in this study was purified by native PAGE, and our ITC analyses show that ~80% of the purified RNA (all conformers except the Y-conformer) is properly folded through its capability to bind ligand.

Page 20. To assess possible concentration-dependent oligomerization, the authors should show 1-D scattering profiles for various riboswitch concentrations over the same momentum transfer range. Here, it is important to show that the thermodynamic ensemble is not changing appreciably in a concentration dependent manner over the concentration range used to create the scattering profile. The reviewer

expects to see that the components used from AFM are an accurate description of the solution ensemble, which is not distorted by concentration dependent aggregation.

We have provided the suggested data in the supplementary information (Supplementary Fig. 3a, page 13 in SI). It is the standard and routine practice to measure a concentration series for multiple purposes.

Page 20. The Guinier plots used to calculate R_g values should be included for transparency and quality control. These plots can also provide confidence that samples are not aggregating (self-associating) or experience radiation damage, which would be apparent in the Guinier analysis.

We have provided the requested data in the supplementary information (Supplementary Fig. 3b, page 13 in SI).

Page 20. Why didn't the authors use the average scattering profile from multiple concentrations for ensemble fitting? What was the rationale for using only 1 μM ? This concentration is significantly higher than the AFM concentration.

The simple average of the scattering profiles, assuming all conformers are equally populated, does not agree with the experimentally measured SAXS data. Instead, a best fit was obtained only by varying the population distribution based on the tally of AFM particles. We have made this clearer in the revised manuscript (page 26 in main text). When the χ^2 between the synthesized SAXS with equal populations of all conformers and the experimental data is ~ 1.57 , worse than 1.02 using population-weighted based on the AFM tally (see the figure below).

The RNA concentrations used in AFM and SAXS are dictated by the technical limitation of each method. The μM concentration is the lowest concentration measurable at the synchrotron with sufficient S/N for analysis, whereas nM concentration is required by AFM for imaging individual molecules to avoid molecular overlap.

Areas that require improved clarity

Page 3. Intro. What do the authors mean by “driving the molecules to uniformity under extreme conditions”?

The meaning of this statement is to emphasize that, although a single stable conformation might be achieved under extreme buffer conditions, namely high Mg^{2+} , as high as 300 mM for crystallization (Johnson, J.E. , et al., 2012), this one conformation is artificially selected and is likely not representative of the conformational landscape that exists in solution, thereby limiting our understanding to the perception of RNA molecules as static structures.

Page 4: It is a little misleading to say, “molecules can be observed in their native states without any manipulation, e.g. labeling, freezing, staining or crystallization” when you immobilize the RNA on a surface. While it is true that you do not need to attach a bulky fluorescent label, the RNA is unnaturally tethered to a surface.

The reviewer’s comment was due to a misunderstanding of the immobilization mechanism (see previous clarification). We reiterate here that the RNA molecules are not mechanically immobilized/tethered through strong covalent interactions, but are rather weakly associated through electrostatic interactions between the mica and RNA molecular surfaces. This type of immobilization has been widely used for more than a decade. Molecules often move on the surface during imaging, and can be removed quite easily through successive washes of the mica surface. As compared to molecules that are labeled, tethered, crystallized under extremely high Mg^{2+} concentrations, etc., but still are often referred to as “native” states or structures, we consider our use of the term “native” to be appropriate.

As a point of clarity. It is unknown what RNA sequence was used for this study. This point must be rectified for the reader. Is the riboswitch the same sequence as that crystallized by Johnson, J.E. , Reyes, F. E., Polaski, J.T. & Batey, R.T. (2012) Nature 492, 133-137 ? If so, this construct contains mutations to aid crystallization (see Supplementary Fig. 1 of that paper). The authors should add a figure or table to the supplement to show the exact sequences transcribed for this work. Does the construct contain the expression platform? This is a transcriptional riboswitch.

The sequence is exactly the same as the one described in Johnson, J.E. , Reyes, F. E., Polaski, J.T. & Batey, R.T. (2012) Nature 492, 133-137, which includes the U1A-binding loop and other modifications to aid in crystallization, and does not contain the expression platform. The sequence information has been added to Methods and the secondary structures have been added in Supplementary Fig. 7a (page 17 in SI).

Page 8, Extended Figure 5. The sequence of the riboswitch cannot be made out due to the small size and low resolution of this figure. Please make this figure clearer so that the reader can understand changes in the sequence associated with M2 and M3 in Fig. 4. A more detailed depiction of the kissing loop (e.g. a stick and ball diagram) would also be helpful to understand the molecular basis for this interaction. This should be called out early in the manuscript. It would help orient the reader to label which portion of the rCbl construct contains the expression platform in supplementary fig. 5. Also Please remove the red squiggly underline in supplementary fig. 5. The purpose is unclear.

We thank the reviewer for these suggestions to improve clarity. We revised Supplementary Fig. 5 (now Supplementary Fig. 7 in the revised manuscript, page 17 in SI) and have added a new Supplementary Fig. 5 (page 15 in SI) showing the kissing-loop interactions in various contexts as well as a call-out in the main

text. The expression platform is not included in this construct.

Fig. 2. It is difficult to see the synthesized SAXS curve, stated to be a red solid line. The line should be moved in front of the black line.

This was an error in the figure caption. The black line is the synthesized SAXS curve. This has been corrected (Fig. 2, page 14 in main text).

Page 15. please provide the company that produces Pico pure water.

Pico pure water refers to the water purified using a Pico Pure Water system (Avidity, UK). We added this information in the revised manuscript (page 20 in main text).

Page 16. What criterion was used to assign the AFM resolution as 5 Å? The definition of resolution should be provided for the reader since most members of the journal audience are unlikely to know. Did samples have an internal reference? Please also state why the resolution is limited by the probe tip.

The AFM resolution does not correspond to the resolution of the atomic structure, but rather the physical resolution of the AFM image, described in Å per pixel. We apologize for this confusion and made the correction in main text (page 21 in main text).

Page 17. The application of 2D FFT and 3rd order polynomial flatten, as well as digital resolution extension appears to be described twice. Was this really the case? See the prior paragraph on page 16.

We thank the reviewer for pointing out this redundancy. The raw images and grayscale images were processed independently using two different software. However, the end results were the same. For simplicity, we have removed the text corresponding to the procedure performed with Gwyddion (page 21 in main text).

Evidence Supports Conclusions?

Page 17. The details of 3-D structure calculations are described in an accompanying paper that is not peer reviewed. This prevents the reader from independently assessing the approach and results. To rectify this shortcoming, the reviewer believes additional Extended information should be presented to describe the approach and the results. In particular, how was the process validated? Why not use proteins (one sequence = one structure) as a benchmark for the methodology? An additional concern, is that it seems plausible that some RNAs will be immobilized on mica to create preferred orientations that would complicate 3-D envelope restorations — like cryo-EM. How is this problem overcome? The methods should also describe the origin of the starting models used for coarse-grained MD. The accompanying methods paper suggests models were used from the PDB. Please clarify.

The starting model used for coarse-grained MD was the PDB structure with missing residues added. We have added a statement in the Methods to clarify this. We have also added a section in the Methods to describe the topological structure calculations and their validation (page 21-23 in the main text). We would like to emphasize that the same approach would not be feasible for proteins where the scale of the structural features in the structural elements, such as alpha-helix and beta-sheet, are not discernable with the current AFM probe technology. This is in contrast to the nucleic acids where the width of major/minor grooves are on a similar scale as the probe, thus discernable. In regards to preferred

orientations, RNA molecules usually are immobilized with a maximum contact surface so that a high-resolution image can be recorded. Contrary to the 3D reconstruction, a single high-resolution image of each particle is sufficient for structure recapitulation. Thus, it is important to note, that this method is quite distinct from 3D-reconstruction from particle averaging, such as that used in cryoEM.

Although it is out of the scope of the current report, we would like to point out clear differences between the recapitulated structure from the single AFM particle image and the crystal structure of the rCbl dimer (see the figure below). While the rmsd of the monomers between the recapitulated and crystal dimers is less than 6 Å, consistent with the estimation in Supplementary Table 3, the rmsd of the dimers between the two structures is 12 Å. This large rmsd between the two structures is due to the helical twist in the AFM structure where no such twist exists in the crystal structure. The “straight” parallel arrangement of the two pairs of P1-P13 helices in the crystal structure can be best explained by crystal packing forces between the domains of symmetry related molecules: L5/L13-L5/L13, P2-P3, and P6 extension. All of these packing interfaces involve the “head” region of the aptamer with a total buried surface area of $\sim 10,000 \text{ \AA}^3$ (10% of the molecular surface) and may prevent the helical twist that is observed in solution by AFM. The difference seen here underscores the importance of determining the structure of individual RNA in solution under relevant physiological conditions.

Figure: Comparison between the AFM and crystal structures of the fully-formed dimers. **A.** The top structure highlighted in a red circle is selected from the E^*CC^N vs. CC plot. The CC score of the top structure is 0.983. **B.** RMSDs between the monomers (left) and dimer (right) of the AFM (green) and the crystal (magenta) structures. The positions of the P1 and P13 helices are indicated. **C.** The side views of the crystal and AFM structures to show the twist in the AFM structure relative to the crystal structure. **D.** Two pairs of arrows show the asymmetry existing in the single particle AFM image, suggesting the helical twist and the asymmetrical arrangement.

Extended Data 1: Why do the AFM and SAXS populations show such large differences in the population contributions of various species? The Y conformation is especially different. This is worth commenting upon since the AFM particles were used to model the SAXS scattering profile.

*We thank the reviewer for pointing this out. The large differences in population were primarily for the Y-conformer and dimer populations. In general, the SAXS dimer populations were consistently higher, most likely owing to the higher concentration (μM) used for SAXS compared to AFM (nM). This point is made clear in the manuscript (page 7 in main text). As for the stark differences in Y-conformer populations, this was most likely due to the difficulty in classifying the particles in this particular image due to limited resolution. To rectify this, we now use the AFM tallies from the higher-resolution images presented in Fig. 1. We have added the following text in the revised manuscript describing the results: "In the absence of ligand, the populations of various conformers derived from SAXS-ensemble fitting show several differences with respect to the particle tallies from AFM images, particularly in the number of monomeric vs. dimeric species (**Fig. 1e and Fig. 2e**). This observation is most likely due to the drastic differences in sample concentrations required for each method. At the low nanomolar concentration for AFM, larger populations of candy and P are observed, with almost no dimer population, whereas the opposite is true for the SAXS-derived populations. These observations suggest that the monomeric species of candy and P can convert to form dimers in a concentration-dependent manner. In the presence of ligand, the dimeric species make up the dominant populations in both methods. Importantly, the Y-conformer population remains relatively unchanged in the absence or presence of ligand, and is independent of rCbl concentration." (page 7 in main text)*

Extended Table 1. What is the basis for the values shown in this table?

Is it from coarse-grained dynamic fitting? Please state this in the table title and provide some explanation of the parameters and their origins.

The values in Supplementary Table 1 are all from the coarse-grained dynamic fitting. This has been made clear in the table title, and detailed explanations are provided in the Methods section (pages 22-23 in the main text).

R_g values in Extended Data Table 3 are inconsistent in terms of significant figures. Also, a precision of 0.1 Å seems more reasonable than 0.01 Å.

We thank the reviewer for the pointing this out and we have revised all supplementary tables accordingly.

Impact of Work/Significance to Field

The Discussion ends somewhat abruptly. It would be helpful to relate the findings to other riboswitches or RNA systems with known heterogeneity problems. For example, the Bsu yvrC cobalamin riboswitch (Chan & Mondrgaon NAR 2020) proved extremely sensitive to folding conditions and required co-transcriptional folding followed by native gel electrophoresis. The work also suggested that the different conformations of the Bsu riboswitch change in response to ligands (e.g., adenosyl cobalamin versus hydroxy or methyl cobalamin). Is it possible that the conformations observed here are responsive to binding other ligands? Did the authors try other ligands besides adenosyl cobalamin to see if they alter the observed solution conformation? Are there other examples of large folded RNAs with rugged folding landscapes that would be amenable to the type of analysis shown here? For example, group I introns or group II introns?

We thank the reviewer for these interesting questions. Since the scope of this study does not include ligand specificity, we have not investigated the effects of different ligands on the conformational landscape. However, we have acquired AFM images on several other large RNAs, including group I intron, all of which show conformational heterogeneity that would be highly intriguing to study further using our approach.

Riboswitches are located within long mRNA transcripts and are widely accepted to act in cis vs in trans. A major finding from this work is that ligand binding induces dimerization with differing modes of contact. However, it is unclear how this finding relates to riboswitch function — which is a driving factor of most structural studies. As the authors know, riboswitches that regulate transcription termination have a narrow temporal window during co-transcriptional folding to affect transcription [Watters, K.E. et al. (2017) Nat. Struct. Mol. Biol. 12, 1124-1131]. Thus, it is unlikely that transcription regulating riboswitches have time (or another RNA with which) to dimerize. While translation regulating riboswitches have a longer window to “switch”, it is unlikely that two mRNAs containing the same riboswitch would find one another inside the packed cell environment to dimerize. Moreover, it has been recently shown that small riboswitches can regulate translation co-transcriptionally [Chatterjee S., Chauvier, A., Dandpat, S.S., Artsimovitch, I. & Walter, N.G. (2020) Proc. Nat. Acad. Sci. U. S. A. 118 e2023426118], wherein the riboswitch is sterically inhibited from dimerizing due to the transcription and translation machinery. The authors are correct that crystal structures of RNAs don't capture flexibility observed in solution experiments and the AFM presented shows distinct oligomers of this RNA in the presence of ligand. However, given previous work, it is unclear how the findings of the current manuscript relate to previously established riboswitch functions in the context of an mRNA. It is also not discussed whether the large changes observed here are representative of all riboswitches, just this riboswitch, or whether the conditions of the AFM experiment exaggerate motions — or represent motions— that may be present inside the crowded folding environment of the cell. These are comments that should be addressed in the authors' rather brief Discussion or a Supplemental Discussion that addresses the larger relevance of the work.

We thank the reviewer for this comment. We make no presumption regarding the biological or functional relevance of dimers/oligomers. We agree with the reviewer that they are highly unlikely to form in vivo. It is noteworthy to point out that the crystal structure by Batey's lab (Johnson, J.E. et al.,

2012, Nature 492, 133-137) is very likely dimeric with the intermolecular kissing loop interaction, not monomeric as described in the paper. This conclusion is based on careful analysis of the electron density map of this structure and modeling of the potential positions of missing residues within the steric constraints of the crystal packing interface (see new Supplementary Fig. 5). A similar dimer that forms through the intermolecular kissing loop interaction was reported for the mesophilic Bacillus subtilis cobalamin riboswitch (Chan C. W. and A. Mondragon, NAR, 48(13), 2020), and is clearly observable in our newly added AFM images of this riboswitch (Supplementary Fig. 2, page 12 in SI).

In regards to the broader implications of this work for other riboswitches and RNA conformational space, generally, we have expanded the discussion section to address these important points (page 11-12 in main text).

Reviewer #2 (Remarks to the Author):

The paper 'Visualizing RNA conformational and architectural heterogeneity in solution' by Jienyu Ding, Yun-Tzai Lee, Yuba Bhandari, Lixin Fan, Charles D. Schwieters, Ping Yu¹, Sergey G. Tarosov, Jason R. Stagno, Buyong Ma, Ruth Nussinov, Alan Rein, Jinwei Zhang, Yun-Xing Wang reports about an AFM imaging study of a single RNA sequence and shows that RNAs are conformationally heterogeneous structures.

Timeliness and importance: X-ray crystallography and, most recently and powerfully, cryo-EM are used to solve structures of biomolecules rather routinely. However, these techniques are not well adapted to study RNA due to their small size and flexibility. Thus, researchers often relied on theoretical RNA structures. Breaking with current concepts, this study shows that RNAs under physiologically relevant conditions are much more diverse than thought. Given the general importance and current emerging interest in RNA molecules, this study is very timely and important.

For the sake of clarity and as a reference system, the authors chose to study RNA cobalamin riboswitches that can equally be studied in crystals and using other methods.

The AFM single molecule data is of excellent quality for which this reviewer congratulates the authors.

The comparison with ensemble SAXS curves is elegant and convincing, and the complementary investigation of binding using ITC gives us confidence in the functional relevance of the findings.

One conformer is evidenced to be inactive and good reason is presented as to why this could be the case.

The paper is clearly and concisely written. I have only one criticism. Despite AFM being the main enabling method in this work, there is literally a single AFM reference (ref 10), which is in my view not even the best chosen. The AFM field has seen a series of improvements in the last decade for the improvement of speed and resolution that merit mention as this paper convincingly proposes AFM as a powerful technique for RNA imaging.

We appreciate the reviewer's comments and we have added more references about AFM in the revised manuscript.

Reviewer #3 (Remarks to the Author):

The manuscript of Ding et al. presents AFM structures of a cobalamin riboswitch. The main finding of the work is that the riboswitch does not adopt a single conformation, but a whole ensemble of them. Based on this, the authors conclude that RNA is much more structurally heterogeneous than previously assumed and that there is no correspondence of one structure per sequence. There is a serious flaw with the assumptions that undermines all the conclusions and makes the manuscript unsuitable for publication. As it is, this work should not be published. The problems described below have been discussed recently in a review by Lennon and Batey (JMB, 2022) on cobalamin riboswitches. The authors should consult this work as it relates to their problems.

It is well known that RNA produced by in vitro transcription followed by purification is prone to be misfolded. There are many possible reasons for this, but a likely explanation is that polymerases commonly used to produce RNA, such as T7 RNA polymerase, are very fast and may pause during transcription. These pauses during in vitro transcription are not the same as the ones that occur using the polymerase that matches the RNA being produced. RNA molecules fold co-transcriptionally, which means that polymerase pauses are crucial to allow some folding events to occur. In vitro RNA folding protocols are also prone to produce misfolded structures unless extreme care is taken and it is not always possible to find a suitable folding protocol. In the case described in the manuscript, the riboswitch is from a thermophilic organism and in vitro transcription was used for RNA production (the polymerase used is not mentioned, but it would typically be T7 RNA polymerase). It is highly likely that the RNA produced is heterogeneously folded due to the production process. Furthermore, many transcriptional riboswitches bind their ligand during transcription and folding of the RNA. Once transcribed, it may not be possible to bind the ligand. As the authors noticed, some of the interactions that may not be present are tertiary interactions, such as kissing loop interaction, which may not be properly formed during transcription if the RNA folds into a conformation that prevents the tertiary interactions to occur. The same RNA was used for crystallographic studies, but in this case the crystallization process itself may serve as a way to select properly folded molecules. It is not surprising that all the experiments are consistent as they all used the same RNA produced in the same manner. It is also not surprising that only one conformation is cobalamin-binding capable, arguing for only one correctly folded state. In the end, it is always important to remember that in vitro observations have to recapitulate in vivo results. In the case of riboswitches, the observation is that in the cell these RNA elements are efficient and work well as on/off switches. There is not a graduated response where only a proportion of them work. They all work. This means that in vivo they are all correctly folded and capable of binding their ligand. This is not what the authors observed using their likely misfolded molecules.

We feel strongly that Reviewer 3's comments do not constitute a fair review of this manuscript and falls far short of the journal's rigorous review standard. Rather than providing a careful analysis of the results, the reviewer refuses to even consider the results by calling into question the legitimacy of the entire study and methodological approach. The basis for this claim about our results, however, is factually incorrect. The reviewer claims that the data we present indicate that "only one conformation is cobalamin-binding capable, arguing for only one correctly folded state." Quite the contrary: our data (AFM tally and ITC data) clearly illustrate that all species except one (Y-conformation) are capable of productive ligand-binding activity. This result is clearly stated in the abstract, illustrated in figures (AFM tallies in Fig 1e; ITC in Fig. 2f, and Fig. 4) as well as a number of places in the text. The reviewer further claims that the majority of conformers we observe are likely misfolded artifacts of in vitro transcribed

RNA, which again, is in contrast to our ITC data and the AFM tally data that show that the majority represents active species.

It is known that co-transcriptional folding in vivo may give a picture of the conformational landscape that might be more relevant to what occurs in cell, but it does not imply by any imagination that only structures studied under crystallization conditions are correctly folded. The reviewer went too far as to argue that in vitro transcribed RNA for structural investigation is unreliable unless it is found in a crystal, thereby essentially invalidating the majority of riboswitch studies reported to date. The reviewer also presumes that, if “in vivo they are all correctly folded and capable of binding their ligand,” then there must be only a single relevant conformation, presumably the crystal structure, and that any other conformations observed in vitro must represent misfolded RNA. This assertion is false and contradicts the fact that multiple conformers are active and functional under the more physiologically relevant solution condition. The reviewer’s claim also contradicts findings reported in the existing research literature. Co-transcriptionally folded aptamers will inevitably form partially folded structures as they are being transcribed. Some of the conformers we observe may indeed represent such partially folded structures along the folding trajectory, but in no way does that imply that they are misfolded or incapable of ligand binding. On the contrary, our data show that despite exhibiting various conformations, the majority of them are active species, demonstrating that these conformers are correctly folded, if by “correctly folded” one means capable of ligand binding, as the reviewer suggests. Moreover, we would also like to point out that the reported crystal structure (Johnson, J.E. et al., 2012, Nature 492, 133-137) was determined under 300 mM Mg²⁺, whereas the intracellular Mg²⁺ concentration is ~1 mM. Such high Mg²⁺ concentration is likely to drive the compaction of the aptamer to a single lowest-energy structure. It has been demonstrated that the structure under high Mg²⁺ concentration is near but distinct from the native structure (Chen et al., NAR, 2012). Furthermore, it is known that btuB cobalamin riboswitch (AdoCbl) adapts multiple conformations under physiologically relevant Mg²⁺ concentration (Choudhary and Sigel, RNA, 2014). The claim that functional RNAs in vivo fold only into a single active conformation is at odds with the general overarching principles about RNA structural dynamics and structure-function relationships. Although in cell structural studies of the cobalamin riboswitch are still lacking, results from in cell studies of other RNAs clearly show heterogeneous RNA conformational landscapes, such as illustrated in well-folded structural elements in HIV-1 genomic RNA (Tomezsko et al., Nature, 2020).

Lastly, since the reviewer appears to claim that crystallization selects only the correct conformer, it is noteworthy to point out that the crystal structure by Batey’s lab (Johnson, J.E. et al., 2012, Nature 492, 133-137) is very likely dimeric with the intermolecular kissing loop interaction, not monomeric as described in the paper. This conclusion is based on careful analysis of the electron density map of this structure and modeling of the potential positions of missing residues within the steric constraints of the crystal packing interface (see new Supplementary Fig. 5). A similar dimer that forms through the intermolecular kissing loop interaction was reported for the mesophilic Bacillus subtilis cobalamin riboswitch (Chan C. W. and A. Mondragon, NAR, 48(13), 2020). As such, we do not see how the dimers we observe by AFM in solution are any less relevant than those observed in crystal.

REVIEWERS' COMMENTS

Reviewer #1 (Remarks to the Author):

Reviewer #1 (Remarks to the Author):

The reviewer has examined the requested marked-up versions the revised manuscript and Supplementary Information. The manuscript is greatly improved with many more details about the approach and validation of the method. The authors were responsive to many of the reviewer's concerns, although the Discussion could be stronger. An excellent control experiment was performed on a mesophilic Cbl riboswitch that alleviates some of the reviewer's concerns about the focus on a thermophilic riboswitch sequence. The SAXS data and rationale are also clearer now. The SAXS analysis provides confidence that the AFM particles represent the major components of the thermodynamic ensemble and are accurately modeled in a manner that recapitulates features of the experimental scattering profile. The reviewers' responses to the authors' answers are highlighted in yellow in an uploaded version or indicated within a series of asterisks *****. New remarks were also added to address new material introduced after the first revision. Responses are given only for passages where the reviewer needed to reply. The reviewer has no significant technical comments.

Noteworthy Results

Fig. 2. The injections should be shown that gave rise to the isotherms in Fig. 2e and elsewhere. This is because there is useful information in the injections that allows the reader to evaluate the quality of the experiment. For example, did the injection return to baseline (i.e., enough time between injections), were there bubbles and a properly adjusted baseline, were there heats of dilution from the ligand, were there buffer mismatches etc. As another point, it is unclear how many times each ITC experiments was repeated? The results should be replicated multiple times (at least 2x) with the same result, including the fractional Napp values.

In the revision, we included the thermogram showing heats of injection in Fig. 2 and additional ITC data in the supplementary information (Supplementary Figure 4, page 14 in SI) to demonstrate the quality of the data and their interpretation. This includes the thermogram for ligand injection into the buffer alone, which was used for baseline subtraction. The experiments were performed a number of times under various conditions.

*****. Please state how many times each experiment was done. A "number of times" is not a rigorous answer. Please include the buffer-subtracted thermogram in the Supplementary Figure 4 panels along with the ligand-to-buffer and ligand-to-receptor titrations. *****.

The reviewer notes that the RNA studied is from *T. tengcongensis*, a thermophilic organism. While the authors varied a several conditions during the folding, they did not attempt analyze the RNA at a

temperature at which this organism is found. A previous study [Marszalkowski et al (2021) RNA 27, 1204-1219] found that in vitro transcribed RNase P RNAs from *Thermus thermophilus* that were incubated at temperature higher than 37 °C had improved catalytic activity. Marszalkowski et al speculated that this high temperature incubation was required because “the native folding transition has an extraordinarily high activation barrier” — likely a trait shared structured RNAs from thermophiles. While RNA from thermophiles usually make suitable candidates for crystallization, due to a high stability at room temperature, it appears to be commonplace to fold the RNA at ≥ 65 °C to ensure folding similar to the native state. Based on this, it seems unlikely that the cobalamin riboswitch from *T. tencongenesis* folds the same at 37 °C solely based on in vitro transcription (presumably) and using a non-*T. tencongenesis* polymerase. This aspect of the approach leaves serious doubts about the folding landscape, which is likely stuck in a number of local minima due to the thermophilic nature of the RNA. This point raises the question of whether the observations here are applicable to mesophilic riboswitches in general. The reviewer recommends conducting a comparable analysis on a mesophilic cobalamin riboswitch for comparison or conducting folding and native purification under conditions better suited to a thermophilic RNA.

Given the last point, the authors should consider the effect of temperature on the AFM studies. As mentioned in the previous point, this RNA is derived from a thermophilic bacterium and likely does not sample all conformations at 25 °C because it has evolved to be functional (i.e. sample these states) at ≥ 65 °C.

The reviewer is correct in pointing out that the RNA used in this study is from a thermophile, and was transcribed with T7 RNA polymerase at 37 °C, purified under non-denaturing conditions, and imaged at 4 °C to maximize image quality and resolution. The reviewer’s point that the folding landscape of this riboswitch under these experimental conditions may in fact be different from that under thermophilic and/or in vivo conditions is also a valid one. Nevertheless, we would like to point out that this study does not aim to delineate (necessarily) a complete ensemble of actual atomic structures associated with the riboswitch under its native cellular conditions, but rather to demonstrate that the single sequence of this RNA is able to fold into heterogeneous conformations, all of which, except for one, are active in terms of ligand binding capability, and thus biologically relevant. From a purely energetic standpoint, we would contend that these structures depict in part the aptamer’s folding landscape and heterogeneous conformations that might be present along the folding trajectory. Again, we emphasize that only one of these conformers (Y-shaped) is energetically trapped (misfolded), as our ITC data show that ~80% of the conformers that exist in solution are capable of ligand binding, and that this process involves conformational changes. It is possible that more conformations exist at elevated temperatures, but this would not alter the intended purpose of this manuscript nor its stated conclusion: “one sequence can fold into multiple (active) conformations.” Given this fact, we do not think that additional experiments at higher temperatures are warranted, nor would they be very practical to conduct on RNA samples at the resolution sufficient for detailed structure analysis, in which high temperature is likely to have negative effects on particle immobilization, imaging quality due to increased motion, and probe stability. To our knowledge, no such experiments have been attempted for high resolution imaging on biological molecules in solution.

To address the reviewer's suggestion of using a mesophilic cobalamin riboswitch, we performed the same experiment using cobalamin riboswitch from *B. subtilis*, a mesophilic strain. As we expected, we observed almost identical behavior of heterogeneous aptamer conformations as seen with that from *T. tengcongensis*. The *B. subtilis* results reaffirm that heterogeneity is not unique to that of *T. tengcongensis* or sample preparation protocols. We provided this new result in the supporting info of the revised manuscript (Supplementary Fig. 2, page 12 in SI).

*****. The new analysis of the mesophilic cobalamin riboswitch complements the existing analysis for the thermophilic riboswitch nicely. This provides confidence that the approach is not the result of an artifact resulting from the thermophile. *****.

Fig 3. Beyond modeling, what other evidence exists that supports the KL as a motif required for heterogeneous dimer formation? In other words, given the untested nature of the authors' approach, additional evidence seems important to support the coarse-grain models and dynamic fitting that forms the basis of this work. What independent evidence supports the modeling approach? This problem appears well suited to single molecule FRET studies or single molecule force extension analysis.

As discussed in the manuscript, we demonstrate by AFM without modeling (Fig. 4a-b, page 17 in main text; Supplementary Fig. 6, page 16 in SI), and the ITC results (Fig. 4d, page 17 in main text; Supplementary Fig. 4c-d, page 14 in SI) the significance of the KL interaction in the proper folding of the aptamer as well as dimer formation. The mutation data show that the inability to form the KL interaction results in ~80% misfolded (Y-conformation) and no indication of dimers, strongly suggesting that dimerization must occur through KL interactions, consistent with our structure analysis. We feel that the complete illustration and validation of the methodology are outside the scope of this manuscript, and will be presented in detail in a separate manuscript to be published elsewhere. smFRET or single molecule force extension experiments are powerful tools, but they are not direct methods and require calibration/verification using structural models. Such experiments are worthy of separate independent studies, but are thus beyond the scope of the current study.

*****. The reviewer believes that the validation of methodology is important to evaluating the reported outcomes of an experimental study. However, the reviewer believes that the SAXS analysis and ability to use the weighted scattering profiles of AFM models to recapitulate the thermodynamic ensemble is strong evidence for the approach. Moreover, the new analysis of the KL interaction in metagenomic rCbl structure (PDB code 4gma) provides support for the intermolecular dimer observed by AFM. This intermolecular dimer was observed in the aCbl structure (PDB code 6vmy). As a point of clarity, however, the labeling of the structure in panel Supplementary Figure 5a as "rCbl" as defined on page 4 of the main text is incorrect since this is not the *Thermoanaerobacter tengcongensis* structure but a marine metagenomic riboswitch structure. Please define it appropriately. This is also true for the following main text on Page 9, ". Investigation of the crystal structure of rCbl26 shows that it may be interpreted as a symmetrical dimer formed through intermolecular KL loop interactions". Again, the crystal structure is not the thermophilic riboswitch but a marine metagenomic variant. *****.

The SI data on the DL are in Supplementary Fig 6. Here the authors provide compelling evidence that the crystal structure from Batey et al. (PDB code 4gma) is actually a dimer at the DL and not the intramolecular model described. This agrees better with Mondragon's model (PDB code 6vmy), which shows an intramolecular dimer. The review looked at the electron density for 4gma and noted breaks in the chain that could have caused ambiguity during modeling this metagenomic riboswitch at 3.94 Å resolution.

The authors state that a major advantage of the approach is that it provides direct visualization of a specific RNA sequence to provide insight into the heterogeneous conformations accessible in solution conditions that approximate the physiological state. However, on page 16 the Methods state that the authors use 1-(3-aminopropyl) silatrane (APS) to immobilize RNA samples on mica. What evidence supports the use of APS as a non-denaturing reagent that preserves the "native" structure of particles immobilized on the mica surface? Since the immobilization depends on covalent bond formation, it is conceivable that the immobilized structures must be partially unfolded. This does not seem to be a true solution state but rather one that is hydrated under controlled conditions. The authors should be more cautious in asserting that the AFM condition is physiologically relevant, especially since the data are based on tapping of the RNA surface with a force probe.

The reviewer's statement of covalent immobilization is factually incorrect. The immobilization is not via covalent alteration of the RNA structure. The functionalization of mica with APS is widely used for the immobilization of nucleic acids primarily through the electrostatic interactions between protonated amino groups of the APS-mica substrate and the negatively charged nucleic acid backbone. The low charge density of APS-mica allows reliable imaging of nucleic acids, protein-nucleic acid complexes and other biological samples, which has been demonstrated in many papers¹⁻³.

1. Shlyakhtenko, L.S. et al. Silatrane-based surface chemistry for immobilization of DNA, protein-DNA complexes and other biological materials. *Ultramicroscopy* 97, 279-87 (2003).
2. Lyubchenko, Y.L., Shlyakhtenko, L.S. & Gall, A.A. Atomic force microscopy imaging and probing of DNA, proteins, and protein DNA complexes: silatrane surface chemistry. *Methods Mol Biol* 543, 337-51 (2009).
3. Stumme-Diers, M.P., Stormberg, T., Sun, Z. & Lyubchenko, Y.L. Probing The Structure And Dynamics Of Nucleosomes Using Atomic Force Microscopy Imaging. *J Vis Exp* (2019).

*****. The reviewer recommends that the authors add a brief discussion of the AFM immobilization conditions and the cited evidence of its use as a non-denaturing approach to probe structure of nucleic acids. This will help to make the findings more broadly accessible to a non-specialized readership. Such a description could go into the Supplementary Information. *****.

Page 15. How confident are the authors that the transcription products are the desired riboswitch? T7 polymerase is known to form untemplated sequences caused by priming of short RNA hairpins as they are transcribed [Nucleic Acids Research, Volume 46, Issue 18, 12 October 2018, Pages 9253–9263]. This suggests that some transcription products purified by native PAGE might not actually be the desired

sequences or folds. The reviewer believes that purified RNAs should be verified by sequencing to prove that the transcripts comprise the desired riboswitch sequence. Next generation sequencing is readily amenable to this problem. The reviewer also believes that there are better approaches for native purification in lieu of gel electrophoresis. For example, Kieft and Batey [RNA 2007 13(8):1384-9] described a native purification method in which the co-transcriptionally folded RNA is affinity purified by MS2 coat protein hairpins and cleaved by a glmS riboswitch. This approach assures that the transcription product has some function and the correct 3'-end sequence for affinity purification. For quality control, the authors should provide evidence that their transcripts are the correct sequence. A denaturing gel with serially diluted RNA is also important to show for quality control of AFM and SAXS experiments.

We have added mass spectrometry data and denaturing-PAGE results in the supporting information to show that the purified RNA represents a correct, homogeneous sequence (Supplementary Fig. 10, page 20 in SI). In regards to the purification method for obtaining functional species, the RNA in this study was purified by native PAGE, and our ITC analyses show that ~80% of the purified RNA (all conformers except the Y-conformer) is properly folded through its capability to bind ligand.

*****. The mass spec and denaturing gel quality control analysis looks excellent and very credible.
*****.

Page 20. To assess possible concentration-dependent oligomerization, the authors should show 1-D scattering profiles for various riboswitch concentrations over the same momentum transfer range. Here, it is important to show that the thermodynamic ensemble is not changing appreciably in a concentration dependent manner over the concentration range used to create the scattering profile. The reviewer expects to see that the components used from AFM are an accurate description of the solution ensemble, which is not distorted by concentration dependent aggregation.

We have provided the suggested data in the supplementary information (Supplementary Fig. 3a, page 13 in SI). It is the standard and routine practice to measure a concentration series for multiple purposes.

*****. The scattering plots look reasonable (not aggregated), although the concentration range shown is small, suggesting that the authors much work in this concentration window. (But see the comments below about concentrations of SAXS and AFM). *****.

Page 20. The Guinier plots used to calculate Rg values should be included for transparency and quality control. These plots can also provide confidence that samples are not aggregating (self-associating) or experience radiation damage, which would be apparent in the Guinier analysis.

We have provided the requested data in the supplementary information (Supplementary Fig. 3b, page 13 in SI).

*****. The Guinier plots at 1 uM look linear, as desired. *****.

Page 20. Why didn't the authors use the average scattering profile from multiple concentrations for ensemble fitting? What was the rationale for using only 1 μM ? This concentration is significantly higher than the AFM concentration.

The simple average of the scattering profiles, assuming all conformers are equally populated, does not agree with the experimentally measured SAXS data. Instead, a best fit was obtained only by varying the population distribution based on the tally of AFM particles. We have made this clearer in the revised manuscript (page 26 in main text). When the χ^2 between the synthesized SAXS with equal populations of all conformers and the experimental data is ~ 1.57 , worse than 1.02 using population-weighted based on the AFM tally (see the figure below).

*****. Why not add this as a Supplemental Figure? *****.

The RNA concentrations used in AFM and SAXS are dictated by the technical limitation of each method. The μM concentration is the lowest concentration measurable at the synchrotron with sufficient S/N for analysis, whereas nM concentration is required by AFM for imaging individual molecules to avoid molecular overlap.

Areas that require improved clarity

Page 3. Intro. What do the authors mean by "driving the molecules to uniformity under extreme conditions"?

The meaning of this statement is to emphasize that, although a single stable conformation might be achieved under extreme buffer conditions, namely high Mg^{2+} , as high as 300 mM for crystallization (Johnson, J.E. , et al., 2012), this one conformation is artificially selected and is likely not representative of the conformational landscape that exists in solution, thereby limiting our understanding to the perception of RNA molecules as static structures.

*****. The authors need to explain this to the reader, not just to the reviewer ... How was the text modified to address this point? *****.

Page 4: It is a little misleading to say, "molecules can be observed in their native states without any manipulation, e.g. labeling, freezing, staining or crystallization" when you immobilize the RNA on a surface. While it is true that you do not need to attach a bulky fluorescent label, the RNA is unnaturally tethered to a surface.

The reviewer's comment was due to a misunderstanding of the immobilization mechanism (see previous clarification). We reiterate here that the RNA molecules are not mechanically immobilized/tethered through strong covalent interactions, but are rather weakly associated through electrostatic interactions between the mica and RNA molecular surfaces. This type of immobilization has been widely used for more than a decade. Molecules often move on the surface during imaging, and can be removed quite

easily through successive washes of the mica surface. As compared to molecules that are labeled, tethered, crystalized under extremely high Mg²⁺ concentrations, etc., but still are often referred to as “native” states or structures, we consider our use of the term “native” to be appropriate.

*****. Again, this level of understanding must be conveyed to the reader, possibly through addition of Supplementary text that helps the authors make their case in this regard. *****.

*****. New Points

Page 12 of the Discussion. The authors describe ligand binding in terms of lock-and-key and induced fit models. However, most riboswitches appear to be somewhere on a pathway between conformational selection and induced fit. [See Suddala et al Walter (2013) NAR 41, 10462, Stoddard et al Batey (2010) Structure 18, 787]. Almost no riboswitches use a lock-and-key model, which implies a perfect fit of the ligand into the binding pocket.

In the second paragraph of the Discussion, the authors perhaps mean “induced [fit] conformational changes”?

In the Discussion, the reviewer also believes it would be appropriate to mention that, “ Overall, we envision the use of AFM in concert with high resolution structural approaches and computational simulation as a new approach to analyze the thermodynamic ensemble of a structured RNA, which provides a more realistic view of the solution conformational states required for biological function”. *****.

Evidence Supports Conclusions?

*****. Yes*****.

Impact of Work/Significance to Field

The Discussion ends somewhat abruptly. It would be helpful to relate the findings to other riboswitches or RNA systems with known heterogeneity problems. For example, the Bsu yvrC cobalamin riboswitch (Chan & Mondrgaon NAR 2020) proved extremely sensitive to folding conditions and required co-transcriptional folding followed by native gel electrophoresis. The work also suggested that the different conformations of the Bsu riboswitch change in response to ligands (e.g., adenosyl cobalamin versus hydroxy or methyl cobalamin). Is it possible that the conformations observed here are responsive to binding other ligands? Did the authors try other ligands besides adenosyl cobalamin to see if they alter the observed solution conformation? Are there other examples of large folded RNAs with rugged folding landscapes that would be amenable to the type of analysis shown here? For example, group I introns or group II introns?

We thank the reviewer for these interesting questions. Since the scope of this study does not include ligand specificity, we have not investigated the effects of different ligands on the conformational

landscape. However, we have acquired AFM images on several other large RNAs, including group I intron, all of which show conformational heterogeneity that would be highly intriguing to study further using our approach.

*****. It is worth mentioning in the manuscript discussion and conclusions that the authors envision using their approach to examine how different ligands elicit different conformational responses. Also, that the approach is amenable to large RNAs. The main point is to broaden the relevance of the findings for the audience and to make the work interesting and accessible to the journal readership – not just responding to the reviewer ... In other words, the reviewer believes it is acceptable to be somewhat speculative about the broader applications of the work here. *****.

Riboswitches are located within long mRNA transcripts and are widely accepted to act in cis vs in trans. A major finding from this work is that ligand binding induces dimerization with differing modes of contact. However, it is unclear how this finding relates to riboswitch function — which is a driving factor of most structural studies. As the authors know, riboswitches that regulate transcription termination have a narrow temporal window during co-transcriptional folding to affect transcription [Watters, K.E. et al. (2017) *Nat. Struct. Mol. Biol.* 12, 1124-1131]. Thus, it is unlikely that transcription regulating riboswitches have time (or another RNA with which) to dimerize. While translation regulating riboswitches have a longer window to “switch”, it is unlikely that two mRNAs containing the same riboswitch would find one another inside the packed cell environment to dimerize. Moreover, it has been recently shown that small riboswitches can regulate translation co-transcriptionally [Chatterjee S., Chauvier, A., Dandpat, S.S., Artsimovitch, I. & Walter, N.G. (2020) *Proc. Nat. Acad. Sci. U. S. A.* 118 e2023426118], wherein the riboswitch is sterically inhibited from dimerizing due to the transcription and translation machinery. The authors are correct that crystal structures of RNAs don't capture flexibility observed in solution experiments and the AFM presented shows distinct oligomers of this RNA in the presence of ligand. However, given previous work, it is unclear how the findings of the current manuscript relate to previously established riboswitch functions in the context of an mRNA. It is also not discussed whether the large changes observed here are representative of all riboswitches, just this riboswitch, or whether the conditions of the AFM experiment exaggerate motions — or represent motions— that may be present inside the crowded folding environment of the cell. These are comments that should be addressed in the authors' rather brief Discussion or a Supplemental Discussion that addresses the larger relevance of the work.

We thank the reviewer for this comment. We make no presumption regarding the biological or functional relevance of dimers/oligomers. We agree with the reviewer that they are highly unlikely to form in vivo. It is noteworthy to point out that the crystal structure by Batey's lab (Johnson, J.E. et al., 2012, *Nature* 492, 133-137) is very likely dimeric with the intermolecular kissing loop interaction, not monomeric as described in the paper. This conclusion is based on careful analysis of the electron density map of this structure and modeling of the potential positions of missing residues within the steric constraints of the crystal packing interface (see new Supplementary Fig. 5). A similar dimer that forms through the intermolecular kissing loop interaction was reported for the mesophilic *Bacillus subtilis* cobalamin riboswitch (Chan C. W. and A. Mondragon, *NAR*, 48(13), 2020), and is clearly observable in

our newly added AFM images of this riboswitch (Supplementary Fig. 2, page 12 in SI).

In regards to the broader implications of this work for other riboswitches and RNA conformational space, generally, we have expanded the discussion section to address these important points (page 11-12 in main text).

*****. This is a very interesting result. The low concentration of the AFM experiments suggests that the dimer equilibrium is more favorable than expected since crystals can be as high as 30 mM in RNA or more. *****.

*****. Methods, page 30. "Rg and intensity at angel zero" should be "angle". *****.

Joseph Wedekind, Ph.D.
Professor
Dept. Biochemistry and Biophysics
University of Rochester School of Medicine & Dentistry
phone: (585) 273-4516

Reviewers' comments:

Reviewer #1 (Remarks to the Author):

The reviewer has examined the requested marked-up versions the revised manuscript and Supplementary Information. The manuscript is greatly improved with many more details about the approach and validation of the method. The authors were responsive to many of the reviewer's concerns, although the Discussion could be stronger. An excellent control experiment was performed on a mesophilic Cbl riboswitch that alleviates some of the reviewer's concerns about the focus on a thermophilic riboswitch sequence. The SAXS data and rationale are also clearer now. The SAXS analysis provides confidence that the AFM particles represent the major components of the thermodynamic ensemble and are accurately modeled in a manner that recapitulates features of the experimental scattering profile. The reviewers' responses to the authors' answers are highlighted in yellow in an uploaded version or indicated within a series of asterisks *****. New remarks were also added to address new material introduced after the first revision. Responses are given only for passages where the reviewer needed to reply. The reviewer has no significant technical comments.

Noteworthy Results

Fig. 2. The injections should be shown that gave rise to the isotherms in Fig. 2e and elsewhere. This is because there is useful information in the injections that allows the reader to evaluate the quality of the experiment. For example, did the injection return to baseline (i.e., enough time between injections), were there bubbles and a properly adjusted baseline, were there heats of dilution from the ligand, were there buffer mismatches etc. As another point, it is unclear how many times each ITC experiments was repeated? The results should be replicated multiple times (at least 2x) with the same result, including the fractional Napp values.

In the revision, we included the thermogram showing heats of injection in Fig. 2 and additional ITC data in the supplementary information (Supplementary Figure 4, page 14 in SI) to demonstrate the quality of the data and their interpretation. This includes the thermogram for ligand injection into the buffer alone, which was used for baseline subtraction. The experiments were performed a number of times under various conditions.

*****. Please state how many times each experiment was done. A “number of times” is not a rigorous answer. Please include the buffer-subtracted thermogram in the Supplementary Figure 4 panels along with the ligand-to-buffer and ligand-to-receptor titrations. *****.

We have added the individual isotherms to accompany the raw thermograms in Supplementary Fig. 4. We have also revised the figure legends for Figure 2 and Supplementary Figure 4 to indicate the number of measurements, which reads as follows: “Titrations with rCbl and M3 were performed twice. The titration with M2 was performed only once since the deletion of P13 is a more extreme approach to abolishing the KL interaction, whose effect is sufficiently demonstrated in M3 which has only disrupting mutations in L13.”

The reviewer notes that the RNA studied is from *T. tengcongenesis*, a thermophilic organism. While the authors varied a several conditions during the folding, they did not attempt analyze the RNA at a temperature at which this organism is found. A previous study [Marszalkowski et al (2021) RNA 27, 1204-1219] found that in vitro transcribed RNase P RNAs from *Thermus thermophilus* that were incubated at temperature higher than 37 oC had improved catalytic activity. Marszalkowski et al speculated that this high temperature incubation was required because “the native folding transition has an extraordinarily high activation barrier” — likely a trait shared structured RNAs from thermophiles. While RNA from thermophiles usually make suitable candidates for crystallization, due to a high stability at room temperature, it appears to be commonplace to fold the RNA at ≥ 65 oC to ensure folding similar to the native state. Based on this, it seems unlikely that the cobalamin riboswitch from *T. tengcongenesis* folds the same at 37 oC solely based on in vitro transcription (presumably) and

using a non-*T. tengcongenesis* polymerase. This aspect of the approach leaves serious doubts about the folding landscape, which is likely stuck in a number of local minima due to the thermophilic nature of the RNA. This point raises the question of whether the observations here are applicable to mesophilic riboswitches in general. The reviewer recommends conducting a comparable analysis on a mesophilic cobalamin riboswitch for comparison or conducting folding and native purification under conditions better suited to a thermophilic RNA.

Given the last point, the authors should consider the effect of temperature on the AFM studies. As mentioned in the previous point, this RNA is derived from a thermophilic bacterium and likely does not sample all conformations at 25 oC because it has evolved to be functional (i.e. sample these states) at ≥ 65 oC.

The reviewer is correct in pointing out that the RNA used in this study is from a thermophile, and was transcribed with T7 RNA polymerase at 37 °C, purified under non-denaturing conditions, and imaged at 4 °C to maximize image quality and resolution. The reviewer's point that the folding landscape of this riboswitch under these experimental conditions may in fact be different from that under thermophilic and/or in vivo conditions is also a valid one. Nevertheless, we would like to point out that this study does not aim to delineate (necessarily) a complete ensemble of actual atomic structures associated with the riboswitch under its native cellular conditions, but rather to demonstrate that the single sequence of this RNA is able to fold into heterogeneous conformations, all of which, except for one, are active in terms of ligand binding capability, and thus biologically relevant. From a purely energetic standpoint, we would contend that these structures depict in part the aptamer's folding landscape and heterogeneous conformations that might be present along the folding trajectory. Again, we emphasize that only one of these conformers (Y-shaped) is energetically trapped (misfolded), as our ITC data show that ~80% of the conformers that exist in solution are capable of ligand binding, and that this process involves conformational changes. It is possible that more conformations exist at elevated temperatures, but this would not alter the intended purpose of this manuscript nor its stated conclusion: "one sequence can fold into multiple (active) conformations." Given this fact, we do not think that additional experiments at higher temperatures are warranted, nor would they be very practical to conduct on RNA samples at the resolution sufficient for detailed structure analysis, in which high temperature is likely to have negative effects on particle immobilization, imaging quality due to increased motion, and probe stability. To our knowledge, no such experiments have been attempted for high resolution imaging on biological molecules in solution.

To address the reviewer's suggestion of using a mesophilic cobalamin riboswitch, we performed the same experiment using cobalamin riboswitch from *B. subtilis*, a mesophilic strain. As we expected, we observed almost identical behavior of heterogeneous aptamer conformations as seen with that from *T. tengcongenesis*. The *B. subtilis* results reaffirm that heterogeneity is not unique to that of *T. tengcongenesis* or sample preparation protocols. We provided this new result in the supporting info of the revised manuscript (Supplementary Fig. 2, page 12 in SI).

*****. The new analysis of the mesophilic cobalamin riboswitch complements the existing analysis fo the thermophilic riboswitch nicely. This provides confidence that the approach is not the result of an artifact resulting from the thermophile. *****.

Fig 3. Beyond modeling, what other evidence exists that supports the KL as a motif required for heterogeneous dimer formation? In other words, given the untested nature of the authors' approach, additional evidence seems important to support the course-grain models and dynamic fitting that forms the basis of this work. What independent evidence supports the modeling approach? This problem appears well suited to single molecule FRET studies or single molecule force extension analysis.

As discussed in the manuscript, we demonstrate by AFM without modeling (Fig. 4a-b, page 17 in main text; Supplementary Fig. 6, page 16 in SI), and the ITC results (Fig. 4d, page 17 in main text;

Supplementary Fig. 4c-d, page 14 in SI) the significance of the KL interaction in the proper folding of the aptamer as well as dimer formation. The mutation data show that the inability to form the KL interaction results in ~80% misfolded (Y-conformation) and no indication of dimers, strongly suggesting that dimerization must occur through KL interactions, consistent with our structure analysis. We feel that the complete illustration and validation of the methodology are outside the scope of this manuscript, and will be presented in detail in a separate manuscript to be published elsewhere. smFRET or single molecule force extension experiments are powerful tools, but they are not direct methods and require calibration/verification using structural models. Such experiments are worthy of separate independent studies, but are thus beyond the scope of the current study.

*****. The reviewer believes that the validation of methodology is important to evaluating the reported outcomes of an experimental study. However, the reviewer believes that the SAXS analysis and ability to use the weighted scattering profiles of AFM models to recapitulate the thermodynamic ensemble is strong evidence for the approach. Moreover, the new analysis of the KL interaction in metagenomic rCbl structure (PDB code 4gma) provides support for the intermolecular dimer observed by AFM. This intermolecular dimer was observed in the aCbl structure (PDB code 6vmy). As a point of clarity, however, the labeling of the structure in panel Supplementary Figure 5a as “rCbl” as defined on page 4 of the main text is incorrect since this is not the *Thermoanaerobacter tengcongensis* structure but a marine metagenomic riboswitch structure. Please define it appropriately. This is also true for the following main text on Page 9, “. Investigation of the crystal structure of rCbl₂₆ shows that it may be interpreted as a symmetrical dimer formed through intermolecular KL loop interactions”. Again, the crystal structure is not the thermophilic riboswitch but a marine metagenomic variant. *****.

We've also noticed that in the PDB database, the organism for this cobalamin riboswitch (PDB ID: 4gma) is incorrectly labeled as “marine metagenome.” However, according to the original paper with which the structure is associated, the sequence of 4gma is indeed from *Thermoanaerobacter tengcongensis*, (Johnson, J.E. et al (2012) Nature 492:133-137) which states it as *TteAdoCbl* riboswitch (PDB ID: 4gma). In addition, the nucleotide sequence matches GenBank AE008691.1 (*Thermoanaerobacter tengcongensis* MB4) (<https://www.ncbi.nlm.nih.gov/nuccore/AE008691.1>).

The SI data on the DL are in Supplementary Fig 6. Here the authors provide compelling evidence that the crystal structure from Batey et al. (PDB code 4gma) is actually a dimer at the DL and not the intramolecular model described. This agrees better with Mondragon's model (PDB code 6vmy), which shows an intramolecular dimer. The review looked at the electron density for 4gma and noted breaks in the chain that could have caused ambiguity during modeling this metagenomic riboswitch at 3.94 Å resolution.

The authors state that a major advantage of the approach is that it provides direct visualization of a specific RNA sequence to provide insight into the heterogeneous conformations accessible in solution conditions that approximate the physiological state. However, on page 16 the Methods state that the authors use 1-(3-aminopropyl) silatrane (APS) to immobilize RNA samples on mica. What evidence supports the use of APS as a non-denaturing reagent that preserves the “native” structure of particles immobilized on the mica surface? Since the immobilization depends on covalent bond formation, it is conceivable that the immobilized structures must be partially unfolded. This does not seem to be a true solution state but rather one that is hydrated under controlled conditions. The authors should be more cautious in asserting that the AFM condition is physiologically relevant, especially since the data are based on tapping of the RNA surface with a force probe.

The reviewer's statement of covalent immobilization is factually incorrect. The immobilization is not via covalent alteration of the RNA structure. The functionalization of mica with APS is widely used for the immobilization of nucleic acids primarily through the electrostatic interactions between protonated

amino groups of the APS-mica substrate and the negatively charged nucleic acid backbone. The low charge density of APS-mica allows reliable imaging of nucleic acids, protein-nucleic acid complexes and other biological samples, which has been demonstrated in many papers¹⁻³.

1. Shlyakhtenko, L.S. et al. Silatrane-based surface chemistry for immobilization of DNA, protein-DNA complexes and other biological materials. *Ultramicroscopy* 97, 279-87 (2003).
2. Lyubchenko, Y.L., Shlyakhtenko, L.S. & Gall, A.A. Atomic force microscopy imaging and probing of DNA, proteins, and protein DNA complexes: silatrane surface chemistry. *Methods Mol Biol* 543, 337-51 (2009).
3. Stumme-Diers, M.P., Stormberg, T., Sun, Z. & Lyubchenko, Y.L. Probing The Structure And Dynamics Of Nucleosomes Using Atomic Force Microscopy Imaging. *J Vis Exp* (2019).

*****. The reviewer recommends that the authors add a brief discussion of the AFM immobilization conditions and the cited evidence of its use as a non-denaturing approach to probe structure of nucleic acids. This will help to make the findings more broadly accessible to a nonspecialized readership. Such a description could go into the Supplementary Information. *****.

We've added the discussion in the Supplementary Information (page 2 in SI).

Page 15. How confident are the authors that the transcription products are the desired riboswitch? T7 polymerase is known to form untemplated sequences caused by priming of short RNA hairpins as they are transcribed [Nucleic Acids Research, Volume 46, Issue 18, 12 October 2018, Pages 9253–9263]. This suggests that some transcription products purified by native PAGE might not actually be the desired sequences or folds. The reviewer believes that purified RNAs should be verified by sequencing to prove that the transcripts comprise the desired riboswitch sequence. Next generation sequencing is readily amenable to this problem. The reviewer also believes that there are better approaches for native purification in lieu of gel electrophoresis. For example, Kieft and Batey [RNA 2007 13(8):1384-9] described a native purification method in which the co-transcriptionally folded RNA is affinity purified by MS2 coat protein hairpins and cleaved by a glmS riboswitch. This approach assures that the transcription product has some function and the correct 3'-end sequence for affinity purification. For quality control, the authors should provide evidence that their transcripts are the correct sequence. A denaturing gel with serially diluted RNA is also important to show for quality control of AFM and SAXS experiments.

We have added mass spectrometry data and denaturing-PAGE results in the supporting information to show that the purified RNA represents a correct, homogeneous sequence (Supplementary Fig. 10, page 20 in SI). In regards to the purification method for obtaining functional species, the RNA in this study was purified by native PAGE, and our ITC analyses show that ~80% of the purified RNA (all conformers except the Y-conformer) is properly folded through its capability to bind ligand.

*****. The mass spec and denaturing gel quality control analysis looks excellent and very credible. *****.

Page 20. To assess possible concentration-dependent oligomerization, the authors should show 1-D scattering profiles for various riboswitch concentrations over the same momentum transfer range. Here, it is important to show that the thermodynamic ensemble is not changing appreciably in a concentration dependent manner over the concentration range used to create the scattering profile. The reviewer expects to see that the components used from AFM are an accurate description of the solution ensemble, which is not distorted by concentration dependent aggregation.

We have provided the suggested data in the supplementary information (Supplementary Fig. 3a, page 13 in SI). It is the standard and routine practice to measure a concentration series for multiple purposes.

*****. The scattering plots look reasonable (not aggregated), although the concentration range shown is small, suggesting that the authors much work in this concentration window. (But see the comments below about concentrations of SAXS and AFM). *****.

Page 20. The Guinier plots used to calculate R_g values should be included for transparency and quality control. These plots can also provide confidence that samples are not aggregating (self-associating) or experience radiation damage, which would be apparent in the Guinier analysis.

We have provided the requested data in the supplementary information (Supplementary Fig. 3b, page 13 in SI).

*****. The Guinier plots at 1 μM look linear, as desired. *****.

Page 20. Why didn't the authors use the average scattering profile from multiple concentrations for ensemble fitting? What was the rationale for using only 1 μM ? This concentration is significantly higher than the AFM concentration.

The simple average of the scattering profiles, assuming all conformers are equally populated, does not agree with the experimentally measured SAXS data. Instead, a best fit was obtained only by varying the population distribution based on the tally of AFM particles. We have made this clearer in the revised manuscript (page 26 in main text). When the χ^2 between the synthesized SAXS with equal populations of all conformers and the experimental data is ~ 1.57 , worse than 1.02 using population weighted based on the AFM tally (see the figure below).

*****. Why not add this as a Supplemental Figure? *****.

We've added this figure in supplementary information (Supplementary Fig. 3c, page 12 in SI).

The RNA concentrations used in AFM and SAXS are dictated by the technical limitation of each method. The μM concentration is the lowest concentration measurable at the synchrotron with sufficient S/N for analysis, whereas nM concentration is required by AFM for imaging individual molecules to avoid molecular overlap.

Areas that require improved clarity

Page 3. Intro. What do the authors mean by "driving the molecules to uniformity under extreme conditions"?

The meaning of this statement is to emphasize that, although a single stable conformation might be achieved under extreme buffer conditions, namely high Mg^{2+} , as high as 300 mM for crystallization (Johnson, J.E. , et al., 2012), this one conformation is artificially selected and is likely not representative of the conformational landscape that exists in solution, thereby limiting our understanding to the perception of RNA molecules as static structures.

*****. The authors need to explain this to the reader, not just to the reviewer ... How was the text modified to address this point? *****.

We've added this point in the introduction (page 3-4 in the main text).

Page 4: It is a little misleading to say, “molecules can be observed in their native states without any manipulation, e.g. labeling, freezing, staining or crystallization” when you immobilize the RNA on a surface. While it is true that you do not need to attach a bulky fluorescent label, the RNA is unnaturally tethered to a surface.

The reviewer’s comment was due to a misunderstanding of the immobilization mechanism (see previous clarification). We reiterate here that the RNA molecules are not mechanically immobilized/tethered through strong covalent interactions, but are rather weakly associated through electrostatic interactions between the mica and RNA molecular surfaces. This type of immobilization has been widely used for more than a decade. Molecules often move on the surface during imaging, and can be removed quite easily through successive washes of the mica surface. As compared to molecules that are labeled, tethered, crystallized under extremely high Mg^{2+} concentrations, etc., but still are often referred to as “native” states or structures, we consider our use of the term “native” to be appropriate.

*****. Again, this level of understanding must be conveyed to the reader, possibly through addition of Supplementary text that helps the authors make their case in this regard. *****.
*****.

We've added this point in the "Immobilization" section in supplementary discussion (page 2 in the SI).

New Points

Page 12 of the Discussion. The authors describe ligand binding in terms of lock-and-key and induced fit models. However, most riboswitches appear to be somewhere on a pathway between conformational selection and induced fit. [See Suddala et al Walter (2013) NAR 41, 10462, Stoddard et al Batey (2010) Structure 18, 787]. Almost no riboswitches use a lock-and-key model, which implies a perfect fit of the ligand into the binding pocket.

In the second paragraph of the Discussion, the authors perhaps mean “induced [fit] conformational changes”?

The reviewer makes a good point regarding the use of “lock-and-key.” We have replaced the phrase with “conformational selection.” We also made the suggested change in the second paragraph to “induced [fit] conformational changes.”

In the Discussion, the reviewer also believes it would be appropriate to mention that, “ Overall, we envision the use of AFM in concert with high resolution structural approaches and computational simulation as a new approach to analyze the thermodynamic ensemble of a structured RNA, which provides a more realistic view of the solution conformational states required for biological function”.

We thank the reviewer for this thoughtful suggestion. We've incorporated it in the end of our Discussion (page 12 in the main text).

*****.

Evidence Supports Conclusions?

*****. Yes*****.

Impact of Work/Significance to Field

The Discussion ends somewhat abruptly. It would be helpful to relate the findings to other riboswitches

or RNA systems with known heterogeneity problems. For example, the Bsu yvrC cobalamin riboswitch (Chan & Mondrgaon NAR 2020) proved extremely sensitive to folding conditions and required co-transcriptional folding followed by native gel electrophoresis. The work also suggested that the different conformations of the Bsu riboswitch change in response to ligands (e.g., adenosyl cobalamin versus hydroxy or methyl cobalamin). Is it possible that the conformations observed here are responsive to binding other ligands? Did the authors try other ligands besides adenosyl cobalamin to see if they alter the observed solution conformation? Are there other examples of large folded RNAs with rugged folding landscapes that would be amenable to the type of analysis shown here? For example, group I introns or group II introns?

We thank the reviewer for these interesting questions. Since the scope of this study does not include ligand specificity, we have not investigated the effects of different ligands on the conformational landscape. However, we have acquired AFM images on several other large RNAs, including group I intron, all of which show conformational heterogeneity that would be highly intriguing to study further using our approach.

*****. It is worth mentioning in the manuscript discussion and conclusions that the authors envision using their approach to examine how different ligands elicit different conformational responses. Also, that the approach is amenable to large RNAs. The main point is to broaden the relevance of the findings for the audience and to make the work interesting and accessible to the journal readership – not just responding to the reviewer ... In other words, the reviewer believes it is acceptable to be somewhat speculative about the broader applications of the work here. *****.

We appreciate the author's suggestion to broaden the scope of the applications of this work. We have added to the Discussion (pp. 11-12), which now includes the very broad applications of understanding RNA folding, ligand selectivity and promiscuity, and the effect of Mg²⁺ concentration on RNA structure and dynamics.

Riboswitches are located within long mRNA transcripts and are widely accepted to act in cis vs in trans. A major finding from this work is that ligand binding induces dimerization with differing modes of contact. However, it is unclear how this finding relates to riboswitch function — which is a driving factor of most structural studies. As the authors know, riboswitches that regulate transcription termination have a narrow temporal window during co-transcriptional folding to affect transcription [Watters, K.E. et al. (2017) Nat. Struct. Mol. Biol. 12, 1124-1131]. Thus, it is unlikely that transcription regulating riboswitches have time (or another RNA with which) to dimerize. While translation regulating riboswitches have a longer window to “switch”, it is unlikely that two mRNAs containing the same riboswitch would find one another inside the packed cell environment to dimerize. Moreover, it has been recently shown that small riboswitches can regulate translation co-transcriptionally [Chatterjee S., Chauvier, A., Dandpat, S.S., Artsimovitch, I. & Walter, N.G. (2020) Proc. Nat. Acad. Sci. U. S. A. 118 e2023426118], wherein the riboswitch is sterically inhibited from dimerizing due to the transcription and translation machinery. The authors are correct that crystal structures of RNAs don't capture flexibility observed in solution experiments and the AFM presented shows distinct oligomers of this RNA in the presence of ligand. However, given previous work, it is unclear how the findings of the current manuscript relate to previously established riboswitch functions in the context of an mRNA. It is also not discussed whether the large changes observed here are representative of all riboswitches, just this riboswitch, or whether the conditions of the AFM experiment exaggerate motions — or represent motions— that may be present inside the crowded folding environment of the cell. These are comments that should be addressed in the authors' rather brief Discussion or a Supplemental Discussion that addresses the larger relevance of the work.

We thank the reviewer for this comment. We make no presumption regarding the biological or

functional relevance of dimers/oligomers. We agree with the reviewer that they are highly unlikely to form in vivo. It is noteworthy to point out that the crystal structure by Batey's lab (Johnson, J.E. et al., 2012, Nature 492, 133-137) is very likely dimeric with the intermolecular kissing loop interaction, not monomeric as described in the paper. This conclusion is based on careful analysis of the electron density map of this structure and modeling of the potential positions of missing residues within the steric constraints of the crystal packing interface (see new Supplementary Fig. 5). A similar dimer that forms through the intermolecular kissing loop interaction was reported for the mesophilic *Bacillus subtilis* cobalamin riboswitch (Chan C. W. and A. Mondragon, NAR, 48(13), 2020), and is clearly observable in our newly added AFM images of this riboswitch (Supplementary Fig. 2, page 12 in SI). In regards to the broader implications of this work for other riboswitches and RNA conformational space, generally, we have expanded the discussion section to address these important points (page 11-12 in main text).

*****. This is a very interesting result. The low concentration of the AFM experiments suggests that the dimer equilibrium is more favorable than expected since crystals can be as high as 30 mM in RNA or more. *****.

*****. Methods, page 30. "Rg and intensity at angel zero" should be "angle". *****.

We've corrected the typo.